# BitLogic: A Framework for Gradient-Based LUT-Native Neural Networks

**Simon Bührer**  *sbuehrer@ethz.ch*
*ETH Zurich*
*Zurich, Switzerland*

**Andreas Plesner**  *aplesner@ethz.ch*
*ETH Zurich*
*Zurich, Switzerland*

**Till Aczel**  *taczel@ethz.ch*
*ETH Zurich*
*Zurich, Switzerland*

**Roger Wattenhofer**  *wattenhofer@ethz.ch*
*ETH Zurich*
*Zurich, Switzerland*

**Reviewed on OpenReview:** *https://openreview.net/forum?id=ZbsSZAfDod*

## Abstract

Gradient-based LUT- and logic-gate-based neural networks (LUTNet, LogicNets, DiffLogic, PolyLUT, NeuraLUT, WARP-LUT, DWN, LILogicNet, LightLUT) replace multiply-accumulate arithmetic with Boolean lookups. The same trained checkpoint deploys to GPU as bitwise ops on bit-packed activations, to FPGA as LUT primitives, and to ASIC as standard-cell gates, all from one code path. Yet each method ships its own training pipeline, encoder, connectivity rule, fan-in, and hardware-reporting convention. The natural practitioner question, which of these choices actually matter for accuracy and which for hardware cost, therefore has no answer in the current literature. We release **BitLogic**, a unified framework that factors the field into a five-axis design space (encoder, connectivity, fan-in, node parameterization, head) and instantiates every prior method under one shared training and evaluation protocol. The framework deliberately omits method-specific procedures such as calibration, pruning, and thresholding, and all evaluations are limited to two-layer feed-forward networks. Combining the per-axis winners identifies a new best-of-space configuration that outperforms every retrained prior on every (dataset, width) cell in which every compared prior fits the shared budget, across MNIST, Fashion-MNIST, CIFAR-10, and CIFAR-100. We evaluate the best-of-space model on all three backends. On MNIST the resulting two-layer network reaches $\sim 126\,\text{MSamples/s}$ on FPGA, $\sim 15\times$ the throughput of a bit-packed GPU forward path that itself processes 64 samples per 64-bit operation, at four-to-five orders of magnitude less energy.

## 1 Introduction

Machine-learning inference now dominates the energy footprint of deployed models. Yang et al. (2024) report that machine-learning workloads accounted for 10 to 15% of Google's total energy use between 2019 and 2021, roughly 60% of which was inference, and Meta reports a 10:20:70 split across experimentation, training, and inference. This has motivated a line of Lookup Table (LUT)-native networks (Wang et al., 2020; Umuroglu et al., 2020; Petersen et al., 2022; Andronic et al., 2025; Andronic & Constantinides, 2025;

Gerlach et al., 2025; Bacellar et al., 2024; Fojcik et al., 2025; Rüttgers et al., 2025) in which each neuron is a small LUT or Boolean gate. The same trained model runs as bitwise ops on a bit-packed Graphics Processing Unit (GPU) forward path, maps directly onto Field-Programmable Gate Array (FPGA) LUT primitives, and synthesizes to standard-cell Application-Specific Integrated Circuit (ASIC) gates, so a single checkpoint covers three deployment backends without retraining.

**The fragmented design space.** The nine gradient-based LUT methods we know of (LUTNet, LogicNets, DiffLogic, PolyLUT, NeuraLUT, WARP-LUT, Differentiable Weightless Neural Network (DWN), LILogic-Net, LightLUT) share the same goal but pick different encoders, connectivity rules, fan-ins, truth-table relaxations, output heads, and hardware-reporting conventions. Their published numbers are therefore not directly comparable. Which choices matter for accuracy and which for hardware cost, and whether any published method sits near the best point of the shared design space, are open questions in the current literature.

**Our approach.** We factor the union of all nine methods into five independent design axes: input encoder, per-layer connection map, per-node Boolean fan-in, node parameterization, and output head. Every prior method becomes one point in this space (Section 3, Table 8). We implement all five axes in **BitLogic** as independently swappable, then sweep each axis one at a time on MNIST under a single shared training protocol (Section 4). Combining the per-axis winners identifies a new best-of-space configuration that no prior paper has trained (Table 8, last row).

We then retrain the six gradient-trained priors with a published weight-learning recipe (DiffLogic, PolyLUT, NeuraLUT, DWN, WARP-LUT, LILogicNet) inside the framework under the same protocol, on MNIST, Fashion-MNIST, CIFAR-10, and CIFAR-100 (Section 6). LUTNet, LogicNets, and LightLUT are placed in the same five-axis map but not independently retrained. Where the shared protocol allows, the retrained columns reproduce each method's reported accuracy. Where it does not, the gap is traceable to method-specific machinery (PolyLUT's structured pruning, DWN's calibration, WARP-LUT's residual block) that the shared protocol disables on purpose. Finally, the best-of-space model deploys to three backends from one code path: a bit-packed GPU path, Vivado post-route on two Xilinx FPGAs, and a target-independent ASIC proxy via Yosys and Nangate 45 nm (Section 7).

**Contributions.**

- The BitLogic framework, released so subsequent work can evaluate a new axis against the whole published slate with one command, with all five axes (encoder, connectivity, fan-in, node parameterization, head) independently swappable (Section 3).
- A protocol-matched cross-method comparison: per-axis sweeps that identify a new best-of-space model (a combination no prior paper has trained, Section 5), and six gradient-trained priors retrained on four standard image-classification benchmarks (Section 6).
- A unified GPU + FPGA + ASIC evaluation of the resulting checkpoint from one code path, with Python and deployed hardware accuracy bit-exact by construction (Section 7).

**Scope.** Empirical claims are limited to feedforward two-layer image classification. Convolutional, residual, attention, and recurrent extensions are deliberately out of scope. The two-layer restriction and the open depth axis are revisited in Section 8.

## 2 Related work

**Weightless neural networks.** Learning over lookup tables and bit-level logic predates the recent FPGA-deployment wave: n-tuple networks (Bledsoe & Browning, 1959) and Wilkie, Stonham and Aleksander's Recognition Device (WiSARD) (Aleksander et al., 1984) store Boolean responses in RAM cells addressed by small input tuples. BTHOWeN (Susskind et al., 2022) bridges the classical weightless line to the modern FPGA-deployment wave with Bloom-filter RAM neurons and a hardware-accelerator design, and its

extended finite-difference gradient estimator is the direct ancestor of the DWN parameterization studied below. Modern differentiable-LUT methods revisit this idea with gradient-based training.

**Gradient-based LUT methods.** Starting with LUTNet (Wang et al., 2020) and LogicNets (Umuroglu et al., 2020), a sequence of methods has proposed different continuous relaxations of Boolean LUT functions, trainable by gradient descent and discretized afterward. Differentiable logic-gate networks (Petersen et al., 2022) relax each node as a softmax over the $2^{2^K}$ Boolean functions of fan-in $K$; the parameter count scales double-exponentially in $K$, so this work is limited to $K{=}2$ in practice. LightLUT (Rüttgers et al., 2025) hits the same node family with a lean Kronecker-indicator basis and soft or straight-through-hard forward sampling, dropping the per-node parameter count to $2^K$. BitLogic instantiates this parametrization at $K \in \{2, 4, 6\}$ and evaluates it head-to-head with the relaxations below. LILogicNet (Fojcik et al., 2025) adds *learnable Top-k sparse routing* on top of the same node family, selecting each node's $K$ inputs from a small candidate pool through a differentiable mask. Other LUT-native relaxations keep the fan-in larger by structuring the truth table: PolyLUT (Andronic & Constantinides, 2023; Andronic et al., 2025) fits a piecewise polynomial and bakes it into the LUT, NeuraLUT (Andronic & Constantinides, 2024; 2025) absorbs a small dense sub-network into each LUT, WARP-LUT (Gerlach et al., 2025) reparameterizes in the Walsh basis, and DWN (Bacellar et al., 2024) keeps the full $2^n$-entry truth table and trains it with an extended finite-difference gradient estimator. The design-space coordinates of the six retrained methods (DiffLogic, PolyLUT, NeuraLUT, DWN, WARP-LUT, LILogicNet) appear in Table 8.

**Orthogonal and complementary directions.** Arithmetic FPGA accelerators such as HLS4ML (Duarte et al., 2018) and FINN / FINN-R (Umuroglu et al., 2017; Blott et al., 2018) target quantized Multiply-Accumulate (MAC) pipelines on Digital Signal Processor (DSP) slices, and LUTMUL (Xie et al., 2025) re-hosts MAC on LUTs; these are complementary to the LUT-native setting of this paper because the underlying primitive is still arithmetic. A survey of LUT-based FPGA Deep Neural Networks (DNNs) is Guo (2025). Topology-focused approaches that vary convolutional and interconnect structure rather than the node itself (Petersen et al., 2024; Kresse et al., 2025), hybrid architectures that combine LUT neurons with conventional arithmetic layers (Nag et al., 2025), and LUT-based models that avoid continuous relaxation entirely such as TreeLUT (Khataei & Bazargan, 2025) are orthogonal to the node-parameterization comparison of this paper and represent natural extensions of the design-space view.

## 3 The five-axis design space

A feedforward LUT network is the composition of five interchangeable components:

$$\underbrace{\mathcal{E}}_{\text{encoder}} \rightarrow \Big[ \underbrace{L_{\mathcal{M}}(f_{\boldsymbol{\theta}}^{(n)})}_{\text{layer: connectivity } \mathcal{M}, \text{ nodes of fan-in } n \text{ and parameterization } f_{\boldsymbol{\theta}}} \Big]^D \rightarrow \underbrace{\mathcal{H}}_{\text{head}}. \tag{1}$$

Composition (1) is read left to right as the forward composition of a single network: a real input vector of dimension $d$ (the number of input features, e.g. 784 for flattened MNIST) is first mapped by the encoder $\mathcal{E}$ to $db$ binary wires, where $b$ is the number of bits emitted per input dimension; the result is then passed through $D$ stacked Boolean logic layers (the $[\,\cdot\,]^D$ bracket denotes $D$ repetitions) and finally aggregated by the head $\mathcal{H}$ into $c$ real-valued class scores. The five axes are (i) the encoder $\mathcal{E} : \mathbb{R}^d \rightarrow \{0, 1\}^{db}$, (ii) the per-layer connection map $\mathcal{M}$, (iii) the per-node Boolean fan-in $n$, (iv) the node parameterization $f_{\boldsymbol{\theta}}$, and (v) the head $\mathcal{H} : \{0, 1\}^w \rightarrow \mathbb{R}^c$. Depth $D$ and per-layer width are architecture-level knobs layered on top. Every prior method in Table 8 fixes a particular setting of these axes. BitLogic exposes all five as independently configurable.

**Nodes are parameterizations, not paradigms.** Each layer holds $w$ Boolean nodes, each implementing an $n$-input Boolean function. Methods differ only in how they relax that discrete object to a differentiable surrogate $f_{\boldsymbol{\theta}}$ during training. The per-method choice is summarized in Table 8. At inference every surrogate discretizes to the same $2^{2^n}$ Boolean truth table, so the deployed LUT format is identical across the field. What differs is which subset of that space is *reachable during training.* PolyLUT at degree $d$ only spans

the $\binom{n}{\leq d}$ Fourier subspace, while NeuraLUT is capped by the per-neuron Multilayer Perceptron (MLP)'s capacity, until the LUT-extraction step rounds each to a full truth table. Parameterizations therefore differ in optimization dynamics and in the corner of the Boolean hypothesis space each can approach, not in the deployed format.

**Connectivity is the main per-layer design choice.** A layer with fan-in $n$ and output width $w$ selects an input tuple per node through a per-node connection mapping $\mathcal{M}_j : \{1, \ldots, n\} \to \{1, \ldots, w_{in}\}$, where $j \in \{1, \ldots, w\}$ indexes the output node (one map per node) and $w_{in}$ is the layer's input width (the encoder output width $db$ for the first layer, the previous layer's $w$ otherwise). Each $\mathcal{M}_j$ assigns the node's $n$ Boolean inputs to wires of the layer input. Two families are common: *fixed* routing, where $\mathcal{M}$ is frozen at construction (random or random-unique initialization) and only the node parameters are learned, and *learnable* routing, where the selection matrix is differentiable during training and snapped to a valid sparse mapping at discretization. The learnable family is further parameterized by a per-slot candidate-pool size, from four candidates up to the full input width (the full-width setting collapses to a matmul fast-path).

**Fan-in is the dominant hardware knob.** The hardware cost of a discrete $n$-input LUT grows as $O(2^n)$ in the worst case. Larger $n$ gives each node more expressive power but more than doubles its LUT footprint. We therefore treat fan-in as a first-class axis of the comparison rather than a node-specific detail.

**Encoders and heads.** The encoder $\mathcal{E}$ maps each continuous input dimension to $b$ bits. We evaluate three families. The *linear thermometer* (Buckman et al., 2018) uses equispaced thresholds, while the *distributive (quantile) thermometer* (Bacellar et al., 2022) uses empirical training-set quantiles. Both emit $b+1$ distinct levels on $b$ wires. The *uniform fixed-point* encoder is a Brevitas-style narrow-range integer quantizer that emits $2^b$ levels on $b$ wires and is the representation used by LogicNets, PolyLUT, and NeuraLUT. The head $\mathcal{H}$ aggregates the final binary feature vector into $c$ real-valued class logits. We use the popcount head studied by Brändle et al. (2025) alongside a quantized variant: a tanh-bounded $c{\times}c$ weight matrix snapped to a signed integer grid that maps onto DSP48 multipliers.

## 4 Experimental protocol

All experiments share one training and evaluation recipe. This single recipe is used throughout, so the cross-method comparison of Section 6 controls for everything except each method's design-space coordinates (Section 3, Table 8) and any structural cap it imposes.

**Architecture.** A two-layer feedforward LUT network (encoder $\to$ logic layer $\to$ logic layer $\to$ head), per Composition (1). Per-layer width is reported per-experiment. Encoder, connectivity, fan-in, node parameterization, and head are the five swept axes and are specified per table.

**Training.** AdamW ($\beta_1{=}0.9$, $\beta_2{=}0.999$, $\epsilon{=}10^{-8}$), constant learning rate $\eta{=}0.01$ with no schedule or warmup, batch size 128, weight decay 0, no label smoothing, no gradient clipping, no mixed precision. Loss is cross-entropy on the raw head logits. Every run trains for 100 epochs with early stopping disabled, two seeds per cell, mean $\pm$ std reported. Per-parameterization initialization, augmentation policy, and seed control are documented in Section A.

**Data.** MNIST, Fashion-MNIST, CIFAR-10, and CIFAR-100 from torchvision. CIFAR training splits are augmented (random horizontal flip, random crop with reflect-mode padding). MNIST and Fashion-MNIST are not. The native training split is divided 90/10 into training and validation via a seed-controlled permutation. The test split is the native torchvision test set. No channel-wise normalization, cutout, or mixup.

**Evaluation.** Every accuracy number we report comes from a bit-packed GPU inference path that evaluates the discretized LUTs with bitwise operations. This path is bit-exact with the emitted SystemVerilog by construction (Section C), so every accuracy number is simultaneously a deployment accuracy. The full hardware pipeline (HDL emission, Vivado post-route, Yosys + Nangate 45 nm) and the definition of each hardware-table column are in Section C.

# 5 Design-space sweep: finding a new best model

We sweep each of the five design-space axes of Section 3 one at a time on MNIST, at three widths per sweep so axis signal is separable from width scaling (full ladder rationale in Section B.1). Every sweep varies one axis at a time on top of a common base architecture: LightLUT (soft) nodes, learnable routing with eight candidates per slot, a quantile thermometer at $b=8$, fan-in $n=4$, and a popcount head. All runs follow the shared protocol of Section 4.

## 5.1 Node parameterization

We sweep the seven node parameterizations of Section 3, listing LightLUT with soft and with straight-through-hard forward sampling as two rows. The gap between the two rows measures the soft-to-hard discretization gap that Yousefi et al. (2025) analyze in detail for differentiable logic-gate networks. All rows use $n=4$ except DiffLogic, which is locked to $n=2$ by construction.

Table 1: Node parameterization sweep on MNIST. Accuracy is mean ± std over two seeds at each width. **NAND2-GE** and **LUTs** report the standard-cell and Vivado-post-route cost of the `layers` submodule at the narrowest width. Smaller is better, bold per column. LightLUT (soft) is the per-width accuracy winner. The top cluster (LightLUT, WarpLUT, DwnLUT) lies within ∼0.7 pp at $w \leq 16$K and tightens to ∼0.3 pp at $w=32$K, and most of the DiffLogic deficit is a fan-in effect, not a node-parameterization one.

| Node | Acc @ 8 K | Acc @ 16 K | Acc @ 32 K | NAND2-GE @ 8 K | LUTs @ 8 K |
|---|---|---|---|---|---|
| LightLUT (soft) | **91.14 ± 0.02** | **94.82 ± 0.02** | **97.11 ± 0.05** | 60,649 ± 164 | 15,242 ± 20 |
| LightLUT (hard) | 91.04 ± 0.06 | 94.62 ± 0.10 | 96.94 ± 0.04 | 61,600 ± 195 | 15,284 ± 17 |
| WarpLUT | 91.03 ± 0.05 | 94.67 ± 0.03 | 96.88 ± 0.05 | 65,542 ± 138 | 15,314 ± 4 |
| PolyLUT | 90.97 ± 0.04 | 94.09 ± 0.00 | 96.53 ± 0.09 | 37,264 ± 257 | 12,358 ± 31 |
| DwnLUT | 90.77 ± 0.02 | 94.50 ± 0.07 | 96.94 ± 0.01 | 60,869 ± 179 | 15,223 ± 26 |
| NeuraLUT | 90.63 ± 0.05 | 93.73 ± 0.19 | 96.22 ± 0.07 | 34,956 ± 459 | 12,064 ± 103 |
| LinearLUT | 90.17 ± 0.04 | 93.39 ± 0.10 | 95.84 ± 0.05 | 33,626 ± 13 | 11,404 ± 4 |
| DiffLogicLUT ($n=2$) | 84.78 ± 0.13 | 89.21 ± 0.03 | 93.53 ± 0.06 | **12,441 ± 164** | **7,138 ± 4** |

Figures 1a and 1b project the same axis onto the cost dimension at matched $n=2$ / $b=4$: each node parameterization is retrained across six widths $w \in \{500, 1{,}000, 2{,}000, 4{,}000, 8{,}000, 16{,}000\}$ and plotted against a Yosys + Nangate 45 nm NAND2-equivalent gate count (Figure 1a) and against a Vivado post-route LUT count on the Alveo U55C (Figure 1b).

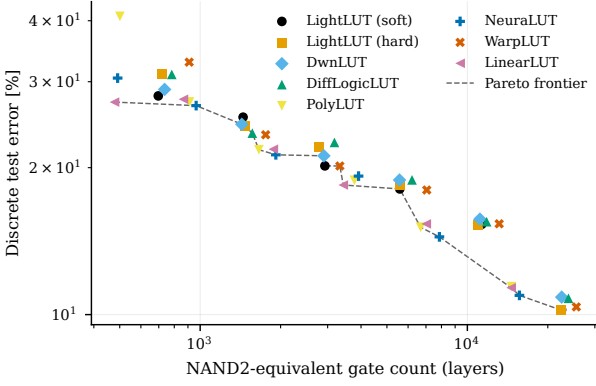 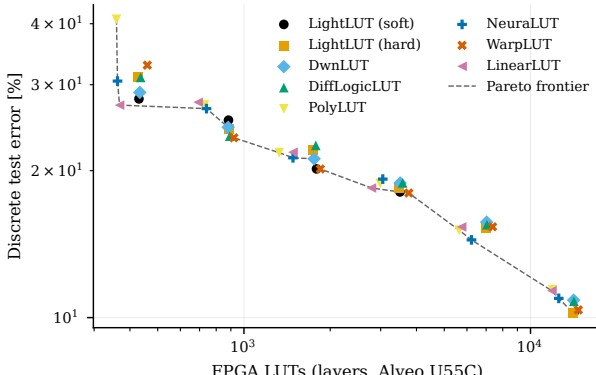

(a) Cost on the `layers` submodule as standard-cell NAND2-equivalent gate count (Yosys + Nangate 45 nm).

(b) Cost as Vivado post-route LUT count of the `layers` submodule on the Alveo U55C target.

Figure 1: MNIST discrete test error versus hardware cost at matched $n=2$ / $b=4$, one series per node parameterization, swept over six widths, both axes log-scaled. Error is evaluated in the HDL-matching forward mode (Section C.2.1), so the reported value equals the deployed-network error rate.

At $w=32K$ the top cluster of Table 1 (LightLUT (soft/hard), WarpLUT, DwnLUT) sits within ∼0.3 pp of each other (96.88–97.11 %), with PolyLUT and NeuraLUT trailing by ∼0.6–0.9 pp. Under matched $n=2$ in Figure 1a the DiffLogic gap to the $n=4$ cluster (of ∼3.6 pp in Table 1) narrows to ∼2 pp, so most of the DiffLogic deficit is a fan-in effect rather than a node-parameterization effect. The per-family error spread narrows with width (from ∼1.5× at the cheap end of the cost sweep to ∼1.1× at the wide end), so the choice of node parameterization matters most under tight hardware budgets.

## 5.2 Connectivity

We sweep six connectivity configurations (Table 2): fixed routing with random or random-unique initialization, and learnable routing at candidate-pool sizes $k \in \{4, 8, 16, \text{full layer}\}$. The ladder is $w \in \{1,000, 2,000, 4,000\}$, narrower than the base ladder because the full-layer learnable cell does not fit at $w{=}8\text{K}$ (Section B.1).

Table 2: Connectivity sweep on MNIST at $w \in \{1,000, 2,000, 4,000\}$. All other axes inherit the base architecture. NAND2-GE and LUTs are reported on the `layers` submodule at the narrowest width. Smaller is better, bold per column. Bounded candidate pools ($k{=}8$–$16$) maximize accuracy. Fixed (random-unique) routing trails the best learnable variant by only $\sim 0.9\,\text{pp}$. Full-layer learnable routing mode-collapses 7–8 pp below every bounded variant. Its NAND2-GE minimum is the hardware shadow of that collapse (see prose and Section B.2), not an efficiency gain.

| Connectivity | Acc @ 1 K | Acc @ 2 K | Acc @ 4 K | NAND2-GE @ 1 K | LUTs @ 1 K |
|---|---|---|---|---|---|
| Learnable, full layer (-1) | $79.10 \pm 0.88$ | $81.18 \pm 0.92$ | $81.47 \pm 0.18$ | $\mathbf{3{,}908 \pm 4}$ | $\mathbf{1{,}262 \pm 4}$ |
| Learnable, 16 candidates | $\mathbf{86.19 \pm 0.27}$ | $\mathbf{87.28 \pm 0.13}$ | $\mathbf{88.84 \pm 0.06}$ | $7{,}325 \pm 212$ | $1{,}850 \pm 5$ |
| Learnable, 8 candidates | $85.02 \pm 0.45$ | $87.22 \pm 0.09$ | $88.51 \pm 0.01$ | $7{,}117 \pm 39$ | $1{,}828 \pm 10$ |
| Learnable, 4 candidates | $85.38 \pm 0.29$ | $86.90 \pm 0.01$ | $88.55 \pm 0.23$ | $7{,}347 \pm 41$ | $1{,}852 \pm 4$ |
| Fixed (random) | $84.55 \pm 0.20$ | $86.95 \pm 0.11$ | $88.19 \pm 0.42$ | $7{,}620 \pm 66$ | $1{,}896 \pm 2$ |
| Fixed (random-unique) | $84.10 \pm 0.10$ | $86.62 \pm 0.16$ | $87.97 \pm 0.01$ | $7{,}755 \pm 29$ | $1{,}917 \pm 9$ |

The full-layer learnable cell is the anomaly of this sweep. It drops to 79.10% at $w{=}1\text{K}$ and 81.47% at $w{=}4\text{K}$, 7–8 pp below every bounded-candidate variant. With no candidate-pool bottleneck the softmax router mode-collapses: many slots converge on the same high-signal wires (a $\sim 20\%$ reduction in distinct input wires per layer between the $k{=}16$ and full-pool configurations, measured from the end-of-training argmax routing). When multiple LUT inputs tie to the same wire, the 16-entry truth table reduces to an effective $\leq 3$-input function and Yosys collapses the gate network accordingly, so the NAND2-GE reduction on that row is a hardware shadow of the same pathology visible in accuracy, not an efficiency gain. Among the non-pathological rows, fixed (random-unique) routing is within $\sim 0.9\,\text{pp}$ of the best learnable variant at $w{=}4\text{K}$ with no candidate-selection overhead.

## 5.3 Fan-in

We sweep $n \in \{2, 4, 6\}$ across the three-width ladder with all other axes at the base architecture (Table 3). Fan-in controls the per-node truth-table size ($2^n$ entries) and is the axis where the standard-cell cost grows fastest, so this sweep maps the accuracy / cost knee that pins the cross-method comparison at $n{=}4$ (Section A).

Table 3: Fan-in sweep on MNIST at three widths. NAND2-GE and LUTs are reported on the `layers` submodule at the narrowest width. Accuracy grows roughly logarithmically with $n$ and the $n{=}2{\to}4$ step is the largest. The standard-cell cost of $n{=}6$ is $\sim 4\times$ that of $n{=}4$. On LUT fabrics $n \leq 6$ maps into a single 6-input LUT, so the ASIC knee is real while the LUT-fabric knee is nearly absent.

| Fan-in | Acc @ 8 K | Acc @ 16 K | Acc @ 32 K | NAND2-GE @ 8 K | LUTs @ 8 K |
|---|---|---|---|---|---|
| $n{=}6$ | $\mathbf{93.83 \pm 0.05}$ | $\mathbf{96.44 \pm 0.01}$ | $\mathbf{97.61 \pm 0.00}$ | $248{,}588 \pm 779$ | $16{,}134$ |
| $n{=}4$ | $91.13 \pm 0.02$ | $94.85 \pm 0.03$ | $97.12 \pm 0.03$ | $60{,}673 \pm 125$ | $15{,}250 \pm 14$ |
| $n{=}2$ | $84.69 \pm 0.08$ | $89.58 \pm 0.11$ | $93.75 \pm 0.06$ | $\mathbf{11{,}358 \pm 192}$ | $\mathbf{7{,}085 \pm 35}$ |

## 5.4 Encoder

We sweep three encoder families, each parameterized by a bit width $b$ (Table 4). The uniform and quantile ("distributive") thermometers are evaluated at $b \in \{4, 8\}$. The binary-coded uniform quantizer at $b \in \{2, 8\}$.

The low-bit point of the binary code isolates the regime where it outperforms the thermometers at half the wire count. $b{=}8$ is common across all three for a matched high-bit comparison.

Table 4: Encoder sweep on MNIST across three widths. All other axes inherit the base architecture. NAND2-GE and LUTs are reported on the `encoder` submodule at the narrowest width. Smaller is better, bold per column. The single "—" cell marks a synthesis job that exceeded its memory budget and was not retried. At matched nominal bit widths the quantile thermometer matches or beats the uniform one by $\sim$0.5 pp on MNIST, but most of that gain and essentially all of the NAND2-GE reduction at $b{=}4$ is an MNIST-specific threshold collapse (Section B.3). The binary-coded quantizer at $b{=}2$ halves the wire count but costs $\sim$4$\times$ more NAND2-GE than the distributive thermometer at $b{=}4$.

| Encoder (bits) | Acc @ 8 K | Acc @ 16 K | Acc @ 32 K | NAND2-GE @ 8 K | LUTs @ 8 K |
|---|---|---|---|---|---|
| `DistributiveThermometer`, $b{=}4$ | $\mathbf{91.26 \pm 0.01}$ | $\mathbf{94.84 \pm 0.07}$ | $\mathbf{97.15 \pm 0.03}$ | $\mathbf{3,397}$ | $\mathbf{1,568}$ |
| `DistributiveThermometer`, $b{=}8$ | $91.15 \pm 0.00$ | $\mathbf{94.84 \pm 0.03}$ | $97.09 \pm 0.01$ | $7,690$ | $3,136$ |
| `Thermometer`, $b{=}4$ | $90.39 \pm 0.05$ | $94.19 \pm 0.12$ | $96.89 \pm 0.09$ | $14,112$ | $5,488$ |
| `Thermometer`, $b{=}8$ | $90.42 \pm 0.17$ | $94.27 \pm 0.08$ | $96.81 \pm 0.09$ | $35,221$ | $10,976$ |
| `UniformFixedPoint`, $b{=}2$ | $89.78 \pm 0.07$ | $93.51 \pm 0.04$ | $96.38 \pm 0.02$ | $13,034$ | $3,920$ |
| `UniformFixedPoint`, $b{=}8$ | $88.59 \pm 0.05$ | $92.61 \pm 0.10$ | $95.55 \pm 0.04$ | — | $12,544$ |

At matched nominal bit widths the quantile thermometer matches or beats the linear one by $\sim$0.5 pp on MNIST (consistent with Bacellar et al. (2022)). Most of the gain and essentially all of the NAND2-GE reduction at $b{=}4$ is an MNIST-specific threshold collapse rather than a real encoder win (Section B.3).

## 5.5 Head

We compare the popcount head with a DSP-backed quantized head (a $c{\times}c$ signed matmul on the length-$c$ group-sum vector with wbits=8, so weights are tanh-bounded during training and snapped to the signed integer grid $[-127, 127]$ at inference, matching the integer MAC the emitter drops into the HDL) across the three-width ladder (Table 5).

Table 5: Head sweep on MNIST across three widths. All other axes inherit the base architecture. NAND2-GE, LUTs, and DSPs are reported on the `head` submodule at the narrowest width. The popcount head is competitive at $w{=}32$K ($-0.7$ pp) with zero DSPs. The DSP-backed head buys $+0.7$–$4.6$ pp across the width ladder at 39 DSPs, $\sim$10% more FPGA LUTs, and $\sim$28% more NAND2-GE: attractive when DSPs are abundant and width is constrained, essentially unjustified at $w{=}32$K.

| Head | Acc @ 8 K | Acc @ 16 K | Acc @ 32 K | NAND2-GE @ 8 K | LUTs @ 8 K | DSPs @ 8 K |
|---|---|---|---|---|---|---|
| `GroupedDSP` | $\mathbf{95.77 \pm 0.33}$ | $\mathbf{97.42 \pm 0.08}$ | $\mathbf{97.81 \pm 0.08}$ | $75,666$ | $12,028 \pm 224$ | $39 \pm 1$ |
| `GroupSum` | $91.14 \pm 0.00$ | $94.84 \pm 0.02$ | $97.12 \pm 0.04$ | $\mathbf{59,080}$ | $\mathbf{10,883}$ | $\mathbf{0}$ |

## 6 Cross-method benchmark

We retrain the six gradient-trained priors (DiffLogic, PolyLUT, NeuraLUT, DWN, WARP-LUT, LILogic-Net) inside the framework under the shared protocol of Section 4, each mapped via Table 8. To keep the comparison fair, the BitLogic best-of-space row pins fan-in to $n{=}4$ and uses the popcount head across all rows. The rationale (and the remaining axis values from Section 5) is in Section A.

Under the shared protocol, the best-of-space configuration wins every retrained (dataset, width) cell in which every compared prior fits the training budget: 62 of the 72 (4 datasets $\times$ 3 widths $\times$ 6 priors) cells in Table 6; the remaining 10 are DWN OOM entries (note [f]) and are not part of that count. At $w{=}64$K the best-of-space reaches 97.84/89.16/58.06/18.82 % on MNIST / F-MNIST / CIFAR-10 / CIFAR-100; the two largest margins are $\sim$1.6 pp over PolyLUT on MNIST and $\sim$5 pp over PolyLUT on CIFAR-10.

Table 6: Best-of-design-space BitLogic against six gradient-trained prior methods retrained inside BitLogic under the common protocol. Accuracy (%) is mean $\pm$ std over two seeds at each width, except the WARP-LUT CIFAR-100 $w$=64K cell, which is single-seed (reported without $\pm$ for that reason). "—" in the Published column means the original paper does not report that (method, dataset) combination. DWN OOM cells are infrastructure-limited by its $O(w^2)$ fast-path and 10-bit thermometer rather than a statement about the method itself (note [f]). Gaps between the Retrained and Published columns reflect the method-specific pipelines the shared protocol disables (Section A), not the design-space coordinates.

| Dataset | Method | Acc @ 4 K | Acc @ 16 K | Acc @ 64 K | Published |
|---------|--------|-----------|-----------|-----------|-----------|
| MNIST | DWN | $87.11 \pm 0.34$ | OOM[f] | OOM[f] | $97.80$–$98.77$[d] |
| | PolyLUT | $86.24 \pm 0.09$ | $91.61 \pm 0.17$ | $96.22 \pm 0.00$ | $96.0$–$97.5$[b] |
| | NeuraLUT | $85.91 \pm 0.33$ | $91.53 \pm 0.11$ | $96.03 \pm 0.09$ | $96.0$[c] |
| | LILogicNet | $80.40 \pm 0.05$ | $88.37 \pm 0.16$ | $95.73 \pm 0.08$ | $97.96$–$98.95$[e] |
| | WARP-LUT | $76.40 \pm 0.22$ | $85.89 \pm 0.10$ | $93.59 \pm 0.04$ | — |
| | DiffLogic | $75.92 \pm 0.30$ | $85.89 \pm 0.03$ | $93.68 \pm 0.05$ | $97.69$–$98.47$[a] |
| | **BitLogic best-of-space** | $\mathbf{88.94 \pm 0.14}$ | $\mathbf{95.05 \pm 0.07}$ | $\mathbf{97.84 \pm 0.04}$ | — |
| F-MNIST | DWN | $78.09 \pm 0.15$ | OOM[f] | OOM[f] | $89.01$–$89.12$[d] |
| | PolyLUT | $74.65 \pm 0.24$ | $83.08 \pm 0.28$ | $87.00 \pm 0.12$ | — |
| | NeuraLUT | $74.27 \pm 0.08$ | $82.46 \pm 0.03$ | $86.48 \pm 0.13$ | — |
| | LILogicNet | $66.50 \pm 0.25$ | $75.54 \pm 0.07$ | $82.10 \pm 0.01$ | — |
| | WARP-LUT | $65.65 \pm 0.38$ | $75.36 \pm 0.12$ | $83.00 \pm 0.05$ | — |
| | DiffLogic | $65.42 \pm 0.05$ | $75.40 \pm 0.04$ | $82.89 \pm 0.12$ | $87.44$[d] |
| | **BitLogic best-of-space** | $\mathbf{78.38 \pm 0.14}$ | $\mathbf{85.90 \pm 0.22}$ | $\mathbf{89.16 \pm 0.08}$ | — |
| CIFAR-10 | PolyLUT | $36.95 \pm 0.02$ | $46.16 \pm 0.01$ | $53.02 \pm 0.16$ | — |
| | NeuraLUT | $36.73 \pm 0.02$ | $44.38 \pm 0.33$ | $47.30 \pm 0.65$ | — |
| | WARP-LUT | $33.86 \pm 0.10$ | $42.92 \pm 0.29$ | $52.12 \pm 0.01$ | — |
| | LILogicNet | $33.83 \pm 0.13$ | $42.36 \pm 0.20$ | $51.67 \pm 0.09$ | $55.11$–$60.98$[e] |
| | DiffLogic | $33.72 \pm 0.16$ | $42.55 \pm 0.02$ | $51.73 \pm 0.34$ | $51.27$–$62.14$[a] |
| | DWN | OOM[f] | OOM[f] | OOM[f] | $57.42$–$57.51$[d] |
| | **BitLogic best-of-space** | $\mathbf{38.93 \pm 0.19}$ | $\mathbf{49.22 \pm 0.26}$ | $\mathbf{58.06 \pm 0.14}$ | — |
| CIFAR-100 | PolyLUT | $8.79 \pm 0.46$ | $12.02 \pm 0.15$ | $16.10 \pm 0.02$ | — |
| | NeuraLUT | $8.71 \pm 0.22$ | $11.88 \pm 0.04$ | $15.20 \pm 0.17$ | — |
| | LILogicNet | $7.63 \pm 0.01$ | $10.62 \pm 0.12$ | $14.54 \pm 0.04$ | — |
| | DiffLogic | $7.49 \pm 0.21$ | $10.61 \pm 0.08$ | $14.64 \pm 0.09$ | — |
| | WARP-LUT | $7.00 \pm 0.04$ | $10.46 \pm 0.00$ | $14.43$ | — |
| | DWN | OOM[f] | OOM[f] | OOM[f] | — |
| | **BitLogic best-of-space** | $\mathbf{10.19 \pm 0.06}$ | $\mathbf{14.06 \pm 0.04}$ | $\mathbf{18.82 \pm 0.09}$ | — |

Published numbers are reproduced from the cited works and are not protocol-matched. They reflect each paper's own training budget and hardware conventions.

[a] DiffLogic: Petersen et al. (2022).

[b] PolyLUT: Andronic & Constantinides (2023); Andronic et al. (2025).

[c] NeuraLUT: Andronic & Constantinides (2024); the published cell quotes only the 2024 NeuraLUT recipe, which is the recipe the retrained row reproduces. NeuraLUT-Assemble 2025 (Andronic & Constantinides, 2025) reports a higher MNIST accuracy of 98.6% but is a successor recipe not reproduced under the shared protocol (Section A).

[d] DWN: Bacellar et al. (2024). The F-MNIST DiffLogic entry is a cross-fill from DWN's Table 1, since F-MNIST is absent from DiffLogic's own paper.

[e] LILogicNet: Fojcik et al. (2025).

[f] DWN's LR-DWN dense connectome uses the full-layer learnable fast-path, whose $O(w^2)$ routing matrix exceeds the training GPU memory budget beyond $w \approx 4{,}000$; only the $w$=4,000 cell is retrained, and the 10-bit thermometer input on CIFAR-10 / CIFAR-100 pushes even that out of budget. LILogicNet is retrained at its paper's flagship Top-32 connectivity and fits the full ladder.

Where the published row exceeds the retrained row by more than $\sim 1\,\mathrm{pp}$ (DWN on MNIST: 97.80–98.77 vs. retrained 87.11; LILogicNet on MNIST: 97.96–98.95 vs. retrained 95.73 at $w$=64K; NeuraLUT-Assemble on MNIST, 98.6, not retrained), the gap traces to a method-specific calibration, pruning, or thresholding pipeline that the shared protocol disables (full list in Section A). Under the shared protocol, the design-space coordinates of the BitLogic best-of-space dominate every retrained prior at every width that fits the training budget.

## 7 Hardware deployment

One checkpoint emits to three backends from a single framework: a bit-packed GPU forward path that processes 64 samples per 64-bit operation, synthesizable SystemVerilog placed and routed by Vivado on two FPGA targets (Alveo U55C, Zynq UltraScale+ XCZU7EV), and a flat Yosys netlist against Nangate 45 nm as a target-independent ASIC proxy. The GPU deployment runs on three consumer cards (Section C). We deploy the MNIST winner of Table 6 at $w$=4,000, the widest cell our build host fits. Wider cells extrapolate linearly at unchanged timing (Section C.5). FPGA numbers are Vivado post-route, with Verilator confirming bit-exact agreement against the Python reference on the full MNIST test set.

**Emission modes.** Each FPGA row picks one of three modes: **max throughput** (parallel encoder/head, fully pipelined, II=1), **lowest latency** (parallel encoder/head, no pipelining), and **fewest resources** (parallel encoder, combinational layers, iterative popcount head over II=$c$ cycles). On the MNIST winner only head iteration pays back. The per-knob breakdown is in Section B.2.

**Cross-platform headline.** Table 7 reports $F_{\max}$, latency, throughput, power, and energy per sample across the three FPGA modes, both targets, and three GPU references at two batch sizes. All rows share the same checkpoint, so accuracy is identical (88.79% on the full MNIST test set, within the two-seed $88.94 \pm 0.14\%$ band of Table 6). Resource footprints are in Tables 11 to 13.

Table 7: Speed and power profile of the MNIST winner at 88.79% test accuracy (single-seed $w$=4,000 checkpoint): three FPGA emission modes on two targets plus three GPU references. **Lat (ns) mixes per-batch and per-sample**: GPU rows report batch wall-clock, FPGA rows per-sample pipeline latency at steady state. Throughput is comparable across platforms, latency is not. Post-route FPGA estimates, not on-board. The bottom row group quotes two published FPGA baselines (Blott et al., 2018; Borras et al., 2022). They are not directly comparable to our rows. Provenance and caveats are in Section C.6. Methodology details in Section C.

| Platform | Mode | $F_{\mathbf{max}}$ (MHz) | Lat (ns) | Tput (kSmp/s) | Power (W) | Energy (nJ/smp) |
|---|---|---|---|---|---|---|
| RTX 3090 | batchsize 64 | 2,115 | 115,712 | 550.0 | 130.67 | 2,287,786 |
| | batchsize 1024 | 2,115 | 116,736 | 8663.7 | 291.42 | 319,425 |
| RTX 2080 Ti | batchsize 64 | 2,100 | 114,224 | 553.5 | 58.02 | 1,014,773 |
| | batchsize 1024 | 2,100 | 146,656 | 6984.6 | 199.60 | 227,080 |
| TITAN RTX | batchsize 64 | 2,100 | 118,784 | 533.0 | 72.49 | 1,273,177 |
| | batchsize 1024 | 2,100 | 135,088 | 7587.7 | 218.91 | 246,347 |
| U55C | max throughput | 126.6 | 40 | 126566.3 | 3.40 | 27 |
| | lowest latency | 84.4 | 12 | 84359.7 | 3.34 | 40 |
| | fewest resources | 110.8 | 99 | 10069.7 | 3.37 | 335 |
| ZU7EV | max throughput | 127.2 | 39 | 127177.9 | 0.72 | 6 |
| | lowest latency | 84.7 | 12 | 84652.5 | 0.66 | 8 |
| | fewest resources | 115.8 | 95 | 10528.0 | 0.68 | 65 |
| *External references* | | | | | | |
| Ultra96 (ZUS+ ZCU3EG) | FINN-R MLP-4 (MNIST) (Blott et al., 2018) | 300.0 | — | 851.7 | 11.80 | 13,856 |
| PYNQ-Z1 (Zynq-7020) | FINN-R MLP-4 (MNIST) (Blott et al., 2018) | 100.0 | — | 162.3 | 2.50 | 15,400 |
| Pynq-Z2 (Zynq-7020) | ResNet-V1 hls4ml (CIFAR-10) (Borras et al., 2022) | — | 27,300,000 | — | — | 44,330,000 |
| Pynq-Z2 (Zynq-7020) | CNV-W1A1 FINN (CIFAR-10) (Borras et al., 2022) | — | 1,500,000 | — | — | 2,535,000 |

The three modes span a clear resource–throughput trade-off: max-throughput buys $\sim1.5\times$ throughput for three orders of magnitude more flip-flops, while *fewest resources* cuts LUTs by $\sim1.5\times$. The pipelined implementation reaches 126.6 MSamp/s (U55C) and 127.2 MSamp/s (XCZU7EV), about $15\times$ the RTX 3090 and $17$–$18\times$ the Turing cards, at 6–27 nJ per sample, four to five orders of magnitude below GPU. The published FPGA numbers in the bottom row group of Table 7 are not a fair comparison, see Section C.6.

## 8 Discussion, limitations, and outlook

**What the sweep taught us.** Two axes give clean signal and two surface structural surprises. *Fan-in* is the single largest lever: the $n{=}2{\rightarrow}4$ step alone explains most of DiffLogic's gap to the $n{=}4$ cluster, and the matched-$n{=}2$ figures (Figures 1a and 1b) narrow that gap from $\sim3.6$ pp to $\sim2$ pp. Once $n$ is fixed, the top *node cluster* (LightLUT, WarpLUT, DwnLUT) converges within $\sim0.3$ pp at $w{=}32$K (Table 1), so the relaxation family matters much less than the literature suggests. *Full-layer learnable connectivity* mode-collapses 7–8 pp below every bounded-candidate variant (Section 5.2). The apparent NAND2-GE win on that row is a hardware shadow of the pathology, not an efficiency gain. The *quantile thermometer*'s lead on MNIST is largely an artefact of an MNIST-specific threshold collapse (Section B.3), not encoder dominance: (pixel > 0) already carries most of the MNIST signal.

**Where the parameterization win shows up.** The per-axis ranking of Table 1 compares node parameterizations at fixed width, and the best-of-space combination of Table 6 amplifies that ranking. On GPU this translates directly into runtime, since cost is set by parameter count and width. Figures 1a and 1b re-project the same axis against ASIC NAND2-equivalent gate count and FPGA LUT count, and the per-family error spread shrinks from about $1.5\times$ at the cheap end of the cost sweep to about $1.1\times$ at the wide end. On those fabric backends the node-parameterization choice is therefore largely orthogonal: any of the node parameterizations sits within about $1.1\times$ of any other at deployment-relevant cost. The accuracy headroom from picking the best-of-space combination matters most when parameter count drives runtime cost, that is, on GPU.

**Depth, the open sixth axis.** Every configuration we evaluate is two layers deep, and we deliberately did not sweep depth. Randomly-initialized LUT stacks lose accuracy rapidly with depth (pilot runs show the soft-vs-hard training gap widening and gradients vanishing through stacked saturating relaxations), and every known remedy introduces a new axis: residual identity initialization (Petersen et al., 2024) and fixed-connectivity routing both mitigate the collapse. We flag depth as the next unsolved axis rather than the next parameter.

**Extensibility.** The five axes are independently swappable at the layer level: a reader with a new node, encoder, or connectivity rule can rerun Section 5 with one config change and measure the new axis against the whole retrained slate. Whole-model topology is fixed in the current release (dense logic layers plus a small set of heads and encoders), but the extension points are clear: convolutional logic layers (Petersen et al., 2024), residual/skip variants, and recurrent LUT layers (Bührer et al., 2025) all exist in isolation. The natural next step is to make each a first-class axis so the same protocol-matched comparison applies. The framework is not bound to image classification, and extending it to other tasks is feasible.

**Outlook.** The pipelined FPGA deployment of Section 7 already runs at sustained throughput high enough that on the edge the sensor and downstream pipeline, not the network, become the bottleneck. Whether this profile extends to attention, residual, or recurrent LUT-native networks at Large Language Model (LLM) scale is the open question BitLogic is meant to enable: every modern neural-network success story (attention, residual learning, convolution, recurrence) has been written in the floating-point idiom, and each needs a LUT-native counterpart before the edge and serving efficiencies reported here transfer to serving-scale workloads. The five axes we swept are the feedforward base case. The next axes are topology, depth, and task.

## 9 Conclusion

Casting every gradient-based feedforward LUT method as a point in one five-axis design space and sweeping each axis under a shared protocol yields a new best-of-space model, a combination no prior paper has trained, that matches or exceeds every retrained baseline on 62 of the 72 (dataset, width, prior) cells of Table 6 across MNIST, Fashion-MNIST, CIFAR-10, and CIFAR-100. The same checkpoint, emitted to bit-exact SystemVerilog and to a standard-cell netlist, reaches the throughput and energy figures reported in Section 7. While this best-of-space accuracy headroom translates into a visible GPU runtime advantage, on FPGA and ASIC the cost-versus-accuracy curves of Figures 1a and 1b cluster across all node parameterizations, so the parameterization choice mainly matters for GPU efficiency.

LUT-based networks push inference onto extremely low-energy hardware, which is broadly positive for the energy footprint of deployed machine-learning systems. Extending the framework beyond image classification is a natural next step.

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

# A    Reproducibility: initialization, prior-method mapping, sweep recipes

This appendix supplements the training protocol of Section 4 with initialization detail, the mapping of each prior method onto the five design-space axes used for the retrained rows of Table 6, the axis constraints that pin those BitLogic rows at $n{=}4$ and the popcount head, the fidelity scope of the retrained column, the per-experiment sweep recipes, and the training-cost numbers. The hardware pipeline (bit-packed inference, HDL emission, Vivado post-route flow, Yosys + Nangate 45 nm standard-cell synthesis) is deferred to Section C.

**Initialization.**    Fixed routing draws its index tensor once per layer as either i.i.d. random or random-unique. Learnable routing draws the candidate-pool mask uniformly and initializes the per-slot routing logits to zero, so the expected selection is uniform across candidates at the first step. Node parameters are drawn from a zero-mean Gaussian (every parametrization's `weight_init="random"` default). The library also exposes an opt-in residual-anchor init that biases each node toward pass-through of its last input, but it is not enabled in any sweep reported here. Quantile-thermometer thresholds are fit once from the empirical pixel quantiles on the training split. The linear thermometer uses equispaced thresholds in the value range.

**Seeds.**    Seeds $\in \{0, 1\}$ drive torch, NumPy, and the 90/10 train/validation split. Every per-axis and cross-method experiment reports mean $\pm$ std over both seeds. The rank-2 node sweep (Figures 1a and 1b) is single-seed by design: it is a cost-vs-accuracy projection rather than a mean comparison, and the within-seed cost spread is already dominated by the width sweep itself.

**Prior-method mapping.**    Each prior method in Table 6 corresponds to the design-space coordinates listed in Table 8 and is trained at three widths $w \in \{4{,}000, 16{,}000, 64{,}000\}$ on each of the four datasets. Two exceptions: DWN's LR-DWN dense connectome uses the full-layer learnable fast-path, whose $O(w^2)$ routing matrix exceeds the GPU memory budget beyond $w{\approx}4{,}000$, so DWN is retrained at $w{=}4{,}000$ only on MNIST and Fashion-MNIST and is OOM at every width on CIFAR-10 / CIFAR-100 (where its 10-bit thermometer further inflates the input tensor). LILogicNet is retrained at its paper's flagship Top-32 connectivity (the 2Top32 variant, Fojcik et al., 2025 Table 3), whose $O(w{\cdot}32)$ routing matrix fits the full ladder. "Learnable($k$)" denotes learnable routing with candidate-pool size $k$, and $k{=}{-}1$ denotes the full-layer fast-path. Encoder family and per-dataset bit count $b$ come from each method's paper. Missing datasets use a nearest-neighbour extrapolation (grayscale $\leftrightarrow$ MNIST, RGB $\leftrightarrow$ CIFAR-10). For DiffLogic, $b{=}1$ on MNIST reproduces the paper's binarized-MNIST single threshold at 0.5, and $b{=}3$ on CIFAR-10 reproduces the "small" row of their Table 5 (fixed per-channel thresholds $\{0.25, 0.5, 0.75\}$). The 31-threshold "large" CIFAR-10 variant is not retrained.

**Axis constraints in the cross-method comparison.**    Two of the five axes are held fixed across all BitLogic rows of Table 6 rather than taken from the per-axis winners of Section 5. **Fan-in is pinned to** $n{=}4$ because (i) the majority of retrained priors (PolyLUT, NeuraLUT, and DWN) natively evaluate at rank 4, and (ii) the fan-in sweep (Table 3) shows accuracy improves faster with width than with rank on the prior-method ladder, so $n{=}4$ keeps most priors at their native rank while retaining width headroom. DiffLogic, WARP-LUT, and LILogicNet, which default to $n{=}2$ in their own papers, therefore sit at rank 4 here as a controlled design-space coordinate, not an accidental mismatch. **The head is pinned to the popcount GroupSum** so the comparison stays pure LUT-only: a DSP-backed head would add a quantized matrix-multiply that no prior method uses, turning the table into a hybrid LUT+DSP vs. LUT-only contest. The remaining three axes (node parameterization, connectivity, encoder) are set to the per-axis winners of Tables 1, 2 and 4.

**Scope of the prior-method reproduction.**    Table 8 aligns each retrained baseline with its paper along the five design-space axes. This is an axis-level alignment, not a full re-implementation. The protocol of Section 4 is shared across all rows, so the comparison is controlled for everything except the design-space choices, and method-specific machinery is intentionally out of scope. PolyLUT-2025's hardware-aware structured-pruning pipeline (dense pre-train with exponential-$\ell_1$ group regularizer, top-$k$ prune, retrain from reinitialized weights, Andronic et al., 2025) and NeuraLUT-Assemble's successor recipe (Andronic & Constantinides, 2025) are not reproduced, so the PolyLUT and NeuraLUT rows use the fixed (random)

sparsity of the 2023 / 2024 originals rather than learned connectivity. DWN's bespoke schedule and quantile-thermometer calibration (Bacellar et al., 2024) are replaced by the shared recipe. LILogicNet's flagship Top-32 learnable connectivity (Fojcik et al., 2025) is kept verbatim, but its dataset-specific binarization thresholds are replaced by linear thermometer thresholds at $i/(b+1)$. WARP-LUT's full residual convolutional block (Gerlach et al., 2025) is replaced by the flat dense backbone used by every other row. The resulting numbers should therefore be read as "each method's parameterization and connection scheme, trained under BitLogic's uniform recipe", not as a reproduction of each headline number.

Table 8: Prior-method to design-space mapping for the retrained rows of Table 6. $n$ is the per-node fan-in, $b$ is the number of thresholds per input feature or colour channel. Superscript † marks a dataset not evaluated in the paper, the value is our closest-analogue extrapolation. The head axis is pinned to the popcount GroupSum across every row (see the axis-constraints paragraph above). The final row is the BitLogic best-of-space configuration identified by the per-axis winners of Section 5, which is trained as a new point in the design space rather than imported from any prior method.

| Method | Node parameterization | Connections | $n$ | Encoder | Encoder bits $b$ | | | |
|---|---|---|---|---|---|---|---|---|
| | | | | | MNIST | F-MNIST | CIFAR-10 | CIFAR-100 |
| DiffLogic | softmax over 16 ops | fixed (random) | 2 | linear thermometer | 1 | 1† | 3 | 3† |
| PolyLUT | degree-$d$ polynomial | fixed (random) | 4 | binary-coded quantizer | 2 | 2† | 2† | 2† |
| NeuraLUT | MLP per neuron | fixed (random) | 4 | binary-coded quantizer | 2 | 2† | 2† | 2† |
| DWN | full truth table (EFD) | learnable($-1$) | 6 | quantile thermometer | 3 | 7 | 10 | 10† |
| WARP-LUT | Walsh basis | fixed (random) | 2 | linear thermometer | 1† | 1† | 3 | 3† |
| LILogicNet | softmax over 16 ops | learnable(32) | 2 | linear thermometer | 1 | 1† | 7 | 7† |
| **BitLogic best-of-space** | LightLUT (soft) | learnable(16) | 4 | quantile thermometer | 4 | 4 | 4 | 4 |

**Per-experiment recipes.** Table 9 lists each paper element with the axis it sweeps, the swept values, and the width ladder. Standard-cell synthesis is not invoked for the per-axis sweeps or the cross-method comparison. Hardware cost is reserved for the rank-2 node figures (Figures 1a and 1b) and the deployment case study (Section 7).

Table 9: Per-experiment sweep configurations behind the tables and figures of Sections 5 and 6. Unless stated otherwise the swept axis is the only deviation from the base architecture of Section 5 (LightLUT (soft), learnable(8), $n=4$, $b=8$ quantile thermometer, popcount head). Every configuration is trained with two seeds. Accuracy is read from the bit-packed inference path on the native test split.

| Element | Swept axis and values | Width ladder |
|---|---|---|
| Table 1 | node: {LightLUT (soft, hard), DwnLUT, PolyLUT, NeuraLUT, WarpLUT, LinearLUT, DiffLogic} | $\{8, 16, 32\}$ K |
| Table 2 | connectivity: {fixed (random), fixed (random-unique), learnable($k$) for $k \in \{4, 8, 16, -1\}$} | $\{1, 2, 4\}$ K |
| Table 3 | fan-in: $n \in \{2, 4, 6\}$ | $\{8, 16, 32\}$ K |
| Table 4 | encoder $\times$ bits: {uniform, quantile}$\times\{4,8\}$, binary$\times\{2,8\}$ | $\{8, 16, 32\}$ K |
| Table 5 | head: {popcount, DSP-backed (wbits=8)} | $\{8, 16, 32\}$ K |
| Table 6 | BitLogic best + 6 prior methods (Table 8), $\times$ 4 datasets | $\{4, 16, 64\}$ K |
| Figures 1a and 1b | node sweep at rank-2 / $b=4$ / learnable(8) / popcount, single seed | $\{0.5, 1, 2, 4, 8, 16\}$ K |
| Table 7–Table 13 | deployment case study driven by the MNIST winner of Table 6 | $w=4$ K |

The BitLogic cells in Table 6 use the best-of-space configuration identified by the per-axis winners of Section 5: LightLUT (soft) nodes, learnable routing with $k=16$ candidates, $n=4$, a $b=4$ distributive thermometer, and a popcount head.

**Training cost.** Table 10 reports per-parameterization wall-clock and peak GPU memory at the largest node-sweep width ($w=32,000$). Training runs on a shared heterogeneous GPU cluster whose scheduler allocates whichever device is free, so seeds of the same cell routinely land on different silicon. Seconds per

epoch scale roughly linearly with width. Within each row the two seeds are grouped by host GPU rather than averaged across dissimilar cards.

Table 10: Training wall-clock, peak GPU memory, and trainable-parameter count per node parameterization at the largest width in the node sweep ($w$=32,000, 2-layer MNIST base). Rows are grouped by host GPU ("/"-separated) rather than averaged across dissimilar silicon. *s / epoch*, *Total (min)*, and *Peak GPU (GB)* mirror that split in the same order, one sub-cell per GPU. Parameter counts are identical across seeds at a fixed configuration.

| Parametrization | GPU host(s) | Trainable params | s / epoch | Total (min) | Peak GPU (GB) |
|---|---|---|---|---|---|
| LightLUT (hard) | RTX 3090 / TITAN RTX | 3,072,000 | 35.50 / 34.07 | 59.16 / 56.78 | 2.535 / 2.535 |
| LightLUT (soft) | RTX 3090 | 3,072,000 | 26.93 | 44.89 | 2.535 |
| DwnLUT | RTX 3090 / TITAN RTX | 3,072,000 | 49.31 / 26.90 | 82.18 / 44.83 | 1.806 / 1.806 |
| DiffLogicLUT ($n$=2) | RTX 3090 / TITAN RTX | 2,048,000 | **15.04** / 19.51 | **25.07** / 32.52 | **0.936** / **0.936** |
| PolyLUT | RTX 2080 Ti | 4,288,000 | 50.28 | 83.80 | 3.320 |
| NeuraLUT | RTX 3090 / TITAN RTX | 5,184,000 | 42.60 / 36.63 | 71.00 / 61.05 | 2.220 / 2.220 |
| WarpLUT | RTX 2080 Ti | 3,072,000 | 44.11 | 73.51 | 2.451 |
| LinearLUT | RTX 3090 | 2,368,000 | 37.47 | 62.45 | 1.852 |

## B  Extended sweep data

This appendix collects the experimental detail that is orthogonal to the narrative of Sections 5 to 7 but that the reader may need when interrogating the headline numbers: why each sweep uses the width ladder it does, which axes are held fixed in the cross-method comparison and why, and which emitter knobs pay back on the MNIST winner in Section 7.

### B.1  Width ladders

The per-axis sweeps are reported at three widths per axis so that axis signal is separable from width scaling. The node, fan-in, encoder, and head sweeps of Section 5 use the base ladder $w \in \{8{,}000, 16{,}000, 32{,}000\}$, centred on the $w$=16,000 base cell. The connectivity sweep uses the smaller ladder $w \in \{1{,}000, 2{,}000, 4{,}000\}$ because its full-layer learnable cell materializes a dense $w \times w$ routing matrix per layer and does not fit at the 8 K base within our training GPU memory budget. The relative ordering of the bounded-candidate rows at that ladder carries over to the wider ladder when re-run, but the full-layer cell is not reachable above $w \approx 4{,}000$ under this recipe. The cross-method comparison of Table 6 uses the wider ladder $w \in \{4{,}000, 16{,}000, 64{,}000\}$ for every row that fits, so the deployment regime is visible. DWN's LR-DWN dense connectome inherits the same full-layer fast-path and is retrained at $w$=4,000 only, with wider cells marked OOM in the same sense.

### B.2  Cost-model asymmetry of the FPGA emitter

The *fewest resources* design point of Section 7 is a selected combination of the emitter knobs, not a blanket "iterate everything" switch. Each knob was measured on the MNIST winner and kept only when it actually reduces whole-model cost.

**Head iteration wins on both cost models.**  Time-sharing the per-class popcount and argmax reduces the head from ~5,170 LUTs (~29.5 k NAND2-eq) to ~2,120 LUTs (~7.6 k NAND2-eq) for $k$=10 classes. The per-class reduction is expensive enough that time-sharing it still wins after paying for the counter and output-latch flip-flops.

**Encoder iteration loses at small input scales.**  The parallel encoder for the MNIST winner is already cheap: $s \cdot b = 784 \cdot 4 = 3{,}136$ thermometer bits, 3,397 NAND2-eq total, because each channel is a short comparator chain. Replacing it with a time-shared comparator over 784 slots requires a 3,136-bit output register whose standard-cell cost alone (~15,680 NAND2-eq at ~5 NAND2 per flip-flop) dominates the

combinational savings, and iterative emission measures $\sim 5.8\times$ worse end-to-end. On FPGA the flip-flops pack into slice positions already bundled with the LUTs, so the register cost is near-zero in slice-LUT accounting, but the input-gather multiplexer still drives the encoder LUT count up from 1,568 to 3,301. At larger input scales the ratio flips, and the MNIST winner is simply too encoder-small for iteration to pay back.

**Layer iteration loses on random-wiring LUT fabrics.** Each LUT's $n$ inputs are independently learned indices into the prior layer's output (no spatial locality, no convolutional structure), so a time-shared layer fabric needs a per-neuron $w$:1 multiplexer on each of its $n$ inputs to switch between the two layers' id tensors. At $w$=4,000 and $n$=4 that gather cost is comparable to the LUT-lookup itself, and adding a 4,000-bit inter-layer register tips the balance. Layer emission therefore stays combinational.

The selected combination (parallel encoder, combinational layers, iterative head) minimizes both FPGA LUT count and Nangate 45 nm gate count on this model. The asymmetry is intrinsic to the interaction between emitter knobs and cost model, not a defect of any one knob. A bigger encoder or a structured connectome would flip these verdicts.

### B.3 MNIST-specific encoder collapse

The NAND2-eq column of Table 4 needs a caveat: the DistributiveThermometer row synthesizes to a quarter of the Thermometer row at matched nominal bit widths ($\sim$3,400 vs. $\sim$14,100 at $b$=4), but these two encoders are not operating at matched *effective* bit widths on MNIST. Quantile thresholds are fit over the globally flattened training pixel distribution, which on MNIST has a point mass at zero exceeding the 88 % quantile. Inspecting the fitted checkpoint: at $b$=4 all four thresholds are 0.0, and at $b$=8 seven of the eight are 0.0 with the last at $\approx 0.706$. The ladder therefore collapses: at $b$=4 every bit is the same (pixel $> 0$) comparison and the encoder is effectively 1-bit, while at $b$=8 it is effectively 2-bit. Yosys hashes the identical per-bit expressions and emits one comparator per pixel. The linear thermometer places its thresholds uniformly in the value range and does not collapse on MNIST, so the two rows compare a genuine $b$-bit code against an effectively 1/2-bit code. That the collapsed variant still leads on accuracy simply confirms that (pixel $> 0$) already carries most of the MNIST signal.

## C Hardware evaluation methodology (GPU, FPGA, ASIC)

This appendix documents how every hardware number in Section 7 is produced: which static-analysis pass populates each column, how the GPU, FPGA, and ASIC paths are made comparable, and how the per-mode cycle budget combines with Vivado's achieved period to yield the reported latency and throughput. A short closing subsection covers the width-scaling extrapolation that licences the single-width hardware report.

No physical FPGA board is programmed for any reported row: the FPGA numbers come from post-place-and-route Vivado reports or, where Vivado's placer bails on wide-port designs, from post-optimization estimates (made precise below). Training runs on a shared heterogeneous GPU cluster (RTX 3090, RTX 2080 Ti, or Titan RTX) under PyTorch 2.x on Compute Unified Device Architecture (CUDA) 12. The exact card, driver, and host CPU per seed are recorded in the run record and unioned into the *GPU host(s)* column of Table 10 at aggregation time.

### C.1 GPU methodology

GPU latency, throughput, and energy are measured against a bit-packed inference path (64 samples per 64-bit operation) that is graph-captured into a single CUDA replay. The remainder of this subsection describes the path's construction, the compiled forward, the timing protocol, the telemetry sampler, and the correctness anchor that ties GPU accuracy to the Python evaluation.

**Bit-packed inference.** The GPU rows of Table 7 and every accuracy number in Sections 5 and 6 use a bit-packed inference path, not the eager forward. At construction the path walks each layer once, extracts the discrete $2^n$-entry truth table plus the per-node routing indices, and normalizes them to a common LUT

input bit order. At inference, 64 samples along the batch dimension are packed into a single 64-bit integer, and each layer's LUT evaluation becomes a bitwise op on packed tensors that processes 64 samples per op. The encoder and head remain in floating point. The packed output is bit-exact with the eager discrete forward and, by construction, with the emitted SystemVerilog (Section C.2.2). On a modern GPU it is roughly one to two orders of magnitude faster than the eager path.

**Compiled forward.** The bit-packed module is wrapped in PyTorch's inference mode and compiled with graph capture, fusing the per-layer pack, bitwise-LUT, and unpack kernels into a single replayable graph. This is the closest GPU analogue of the emitted SystemVerilog: one submitted unit of work per sample batch, no per-iteration driver round-trips. The batch is moved to the device once before timing, so no host-to-device transfers and no dataloader work enter the timed region.

**Timing protocol.** Per-iteration latency is measured with CUDA events placed as barriers between back-to-back forwards. Warmup fires the forward without host synchronization until both an iteration-count floor (500 iters) and a wall-clock floor (5 s) are met, long enough for the card's Dynamic Voltage and Frequency Scaling (DVFS) controller to settle out of its idle P-state, which matters on Turing consumer silicon. A single device synchronization then gates entry to the timed region, where the driver records $N+1$ barrier events and submits $N=5{,}000$ back-to-back forwards. Per-iteration latency is recovered from the event-to-event elapsed time. Throughput is $N \cdot$ batch divided by a wall-clock bracket around the loop. Each device is benchmarked at two batch sizes (64 and 1024) so small-batch and sustained regimes are both visible.

**Energy and clock telemetry.** A background NVIDIA Management Library (NVML) sampler polls power, SM clock, memory clock, and junction temperature at 10 Hz over the timed region. Power samples are integrated to joules with the trapezoidal rule. Clocks and temperature are summarized as min/mean/max. The card's nominal boost ceiling is read once at startup, and a row whose observed peak SM clock sits more than 10% below that ceiling is flagged as DVFS-limited rather than compute-bound, so an artificially depressed clock is not silently read as a latency win.

**Correctness anchor.** Before timing, the driver compares a one-batch eager forward against the bit-packed forward: exact equality for LightLUT checkpoints, a $10^{-4}$ element-wise tolerance for softer parameterizations. Test accuracy through the bit-packed path is recorded alongside the timing numbers, so every GPU row carries its own correctness check and is bit-exact with the Python accuracy reported by Sections 5 and 6.

Table 11: Peak GPU memory (weights plus activations) for the MNIST winner at two batch sizes. Model-dependent, not card-dependent: the per-card values coincide, so one multirow lists all three benchmarked cards.

| Platform | Mode | Memory (MB) |
|---|---|---|
| RTX 3090 / RTX 2080 Ti / TITAN RTX | batchsize 64 | 7.27 |
| | batchsize 1024 | 10.15 |

## C.2 FPGA methodology

The HDL emitter writes one synthesizable SystemVerilog top module per emission mode and records its cycle budget $(D, \mathrm{II}, C)$, i.e. pipeline depth, initiation interval, and cycles-per-sample. Vivado then turns those constants and the achieved clock period into the latency, throughput, and resource numbers reported in Tables 7 and 12.

### C.2.1 HDL export and emission modes

Every FPGA number originates from a single trained checkpoint. The HDL emitter extracts each layer's truth table and routing indices and writes synthesizable SystemVerilog. The top module inlines the full forward path (encoder comparators, logic layers, and head) as a single block, with quantized encoder code

as input and predicted class index as output, so every downstream resource number covers the whole model end-to-end.

Three orthogonal emitter options select the three modes of Section 7: encoder style (parallel vs. iterative), head style (parallel vs. iterative), and three independent pipeline flags (encoder / logic layers / head). Each mode fixes a deterministic cycle budget $(D, \mathrm{II}, C)$ that the emitter records before any synthesis. For an $L$-layer stack, the core pipeline depth is

$$D_{\mathrm{core}} = \not\!\mathrm{F}_{\mathrm{pipe\_enc}} + L \cdot \not\!\mathrm{F}_{\mathrm{pipe\_layers}} + \not\!\mathrm{F}_{\mathrm{pipe\_head}}.$$

- **Max throughput.** Parallel encoder and head, with pipeline flip-flops at the encoder, every logic layer, and the head. $D_{\mathrm{core}}{=}4$ on the two-layer MNIST winner, with II=1 (one sample in, one sample out per cycle).
- **Lowest latency.** Parallel encoder and head, pipelining off, so the core datapath is purely combinational ($D_{\mathrm{core}}{=}0$). The shim wraps it as a single flop-to-flop stage for Vivado.
- **Fewest resources.** Parallel encoder, combinational logic layers, iterative popcount head. A single shared popcount/argmax updater walks the $c$ output classes sequentially, so $\mathrm{II}_{\mathrm{core}}{=}c$ ($c{=}10$ on MNIST). Iterating the encoder or layers is a net regression on random-wiring LUT networks at MNIST scale (Section B.2).

The full-module rows additionally enable the Block Random Access Memory (BRAM)-backed I/O shim (Section C.2.3), whose output register adds exactly one flop-to-flop stage. The cycle-budget calculator folds that +1 into $D$, II, and $C$ when the shim is enabled, so the table renderer needs no shim-specific branch.

### C.2.2 HDL fidelity

For every checkpoint used in Section 7 we run an equivalence check on the full test split: Verilator compiles the emitted SystemVerilog with an auto-generated byte-streaming harness, each test sample is piped through the binary, and its one-byte class-index output is compared against the bit-packed inference argmax. All rows across all three emission modes on both FPGA targets report agreement 1.0 on the full test set, certifying that every FPGA accuracy number is bit-exact with the reported Python evaluation. The same check runs per seed in the per-axis sweeps on a 1,000-sample subsample for speed.

### C.2.3 Vivado deployment

The two FPGA targets, an Alveo U55C (part `xcu55c-fsvh2892-2L-e`) and a Zynq UltraScale+ XCZU7EV (part `xczu7ev-ffvc1156-2-e`), share the UltraScale+ 6-LUT primitive (so resource counts are directly comparable) and differ in capacity ($\sim$1.3 M LUTs on U55C vs. $\sim$230 k on the XCZU7EV). Both run the same non-project Vivado 2024.2 flow, synthesizing out-of-context to skip I/O buffer insertion.

**BRAM-backed I/O shim.** The MNIST winner's $784 \times 8 = 6{,}272$-bit encoder input does not fit either part: the count is far above any Xilinx package's pin budget and trips Vivado's placer IO-rule check before placement even begins. To make the design synthesizable *and* make the reported numbers reflect real evaluation rather than constant-folded ghosts, the emitter wraps the core in a thin registered shim with a small pin footprint (clock, reset, class-id). A four-entry BRAM-inferred Read-Only Memory (ROM) holds distinct sample bit patterns. On every cycle the next sample is buffered into a wide input register driving the core, and the core's output is captured one cycle later into an output register. Because real, varying sample data is clocked through the network each cycle, Vivado sees genuine flop-to-flop datapaths to analyse. It cannot constant-fold the LUTs into a fixed output, and routed timing and power are reported against meaningful switching activity. The shim is a synthesis harness, not a deployment interface. A real system would stream samples through the same registered input port over AXI-Stream or High-Bandwidth Memory (HBM), with sample-transport latency additive to the reported numbers. The shim's output register adds exactly one flop-to-flop stage on top of the core's cycle budget. The emitter folds that +1 into $D$, II, and $C$ when the shim is enabled, so the formulas in Section C.4 stay shim-agnostic. Only the full-module row uses the shim. The three submodule runs (encoder, logic-layer stack, head) stop after optimization, since Table 12 only consumes their LUT/Flip-Flop (FF) counts.

**Clock constraint.** Every run that enters timing analysis is constrained against a real clock (the shim always exposes a clock port), with mode-specific target periods picked so Vivado meets timing post-route:

| Mode | $T_{\mathrm{clk}}$ (ns) | Rationale |
|---|---|---|
| Max throughput | 8 | Pipelined, II=1; achieved critical path $\approx$7.4–7.7 ns on both parts, so 8 ns leaves 4–8% margin. |
| Lowest latency | 12 | Single combinational pass through the full forward. |
| Fewest resources | 10 | Combinational datapath with an iterative head Finite-State Machine (FSM); per-cycle path is short. |

The actual $T_{\mathrm{clk}}$ is recorded per run so the achieved period is reconstructed as $T = T_{\mathrm{clk}} - \mathrm{WNS}$.

**Power.** Power is produced by Vivado's vectorless activity estimation on the full-module routed netlist (submodule runs skip the power report). A switching-activity-driven flow would tighten Vivado's "Medium"-confidence default and is left to future work.

Table 12: Vivado post-route (**model**) and post-opt (**encoder**, **layers**, **head**) LUT and flip-flop utilization per emission mode on the two FPGA targets. The **model** column reports the whole synthesized design and is *not* the arithmetic sum of the submodule columns, because Vivado's synthesizer shares logic across module boundaries.

| Platform | Mode | model LUT | model FF | encoder LUT | encoder FF | layers LUT | layers FF | head LUT | head FF |
|---|---|---|---|---|---|---|---|---|---|
| U55C | max throughput | 12,515 | 7,857 | 1,568 | 784 | 7,157 | 7,844 | 5,169 | 4 |
| | lowest latency | 11,977 | 6 | 1,568 | 0 | 7,461 | 0 | 5,170 | 0 |
| | fewest resources | 8,123 | 38 | 1,568 | 0 | 7,461 | 0 | 2,121 | 40 |
| ZU7EV | max throughput | 12,478 | 7,864 | 1,568 | 784 | 7,157 | 7,844 | 5,169 | 4 |
| | lowest latency | 11,818 | 6 | 1,568 | 0 | 7,461 | 0 | 5,170 | 0 |
| | fewest resources | 7,870 | 38 | 1,568 | 0 | 7,461 | 0 | 2,121 | 40 |

## C.3 ASIC methodology

The target-independent NAND2-equivalent gate count in Table 13 comes from a Yosys-driven flow against the Nangate 45 nm open-cell library: read SystemVerilog, flatten, technology-map against the Liberty file, and sum post-map cell counts weighted by cell area over the NAND2 reference cell area. The resulting NAND2-equivalent count is a technology-neutral proxy that lets a reader cross-check against published numbers from other logic-based DNN frameworks. We deliberately do not attach a gate count to the per-axis sweep tables of Section 5: every axis is swept at three widths, and an NAND2-equivalent column would be dominated by the linear width scaling and bury the axis signal.

Table 13: Target-independent NAND2-equivalent gate count from Yosys + Nangate 45 nm standard-cell synthesis, per emission mode, over the four columns **model**, **encoder**, **layers**, **head**. The **model** column reports the whole synthesized design and is *not* the arithmetic sum of the submodule columns, because the area mapper re-balances logic cones across module boundaries.

| Platform | Mode | model | encoder | layers | head |
|---|---|---|---|---|---|
| Nangate 45 nm | max throughput | 109,073 | 7,840 | 72,623 | 29,471 |
| | lowest latency | 63,616 | 3,397 | 28,154 | 29,549 |
| | fewest resources | 40,004 | 3,397 | 28,154 | 7,624 |

## C.4 How each number is obtained

Each numeric cell of Tables 7 and 11 to 13 is the product of a single static-analysis pass and a deterministic arithmetic derivation. Writing $T = T_{\mathrm{clk}} - \mathrm{WNS}$ for the achieved period and $(D, \mathrm{II}, C)$ for the emitter's cycle budget (on shim-enabled rows these include the +1 flop-to-flop stage from the shim's output register):

- **$F_{\max}$ (MHz).** $F_{\max} = 1000/T$. Identical across all three modes. WNS is the post-route signed slack against the mode's constraint.
- **Latency (ns).** Mode-dependent. With the shim every mode reduces to a flop-to-flop chain of length $C$, so latency is uniformly $C \cdot T$: $T$ for lowest-latency ($C{=}1$); $D \cdot T$ for max-throughput ($C{=}D$); $C \cdot T$ for fewest-resources (each sample occupies the shared FSM for $C$ cycles including the output register).
- **Throughput (kSmp/s).** Tput $= 10^3 \cdot F_{\max}/$II, with II=1 for lowest-latency and max-throughput and II=$C$ for fewest-resources.
- **Power (W).** Full-module Vivado vectorless estimation on the routed netlist. Submodule runs do not produce a power cell.
- **Energy/smp (nJ).** $E = 10^9 \cdot P/(10^3 \cdot \text{Tput})$, i.e. watts divided by samples-per-second, expressed in nJ.
- **LUT / FF.** Vivado utilization report, emitted after optimization and overwritten by the post-route counts when place-and-route completes. DSP and BRAM are zero for the MNIST winner by construction (fabric-only) and are omitted from Table 12.
- **NAND2-eq.** Yosys + Nangate 45 nm flow above, target-independent, counted over the flattened top module so encoder comparators and the head output stage are included alongside the logic layers.

The GPU rows of Table 7 use the same single-sample latency (wall-clock through the bit-packed path) and throughput definitions as the FPGA rows, so every row in the table is directly comparable.

## C.5 Width scaling of hardware metrics

Resource and power scale linearly in layer width $w$ with a width-independent offset, while timing is width-invariant to first order. The single-width hardware report therefore extends in closed form. The deployment case study reports a single layer width ($w{=}4{,}000$), the largest cell of Table 6 that the Vivado synthesis flow admits within our build-host memory budget on the emitted flat SystemVerilog. The numbers extrapolate to wider cells of the same two-layer MNIST architecture in closed form: each logic-layer node emits exactly one $2^n$-entry LUT in the SystemVerilog (one `localparam` truth table plus one indexed `assign` per output neuron), so resource and power counts are linear in $w$ with a width-independent offset. Writing $L$ for the number of logic layers ($L{=}2$ on MNIST), $c$ for the number of classes, and holding fan-in $n$, depth $L$, and emission mode constant:

$$\text{LUT}(w) \approx \alpha_{\text{LUT}} \cdot L \cdot w + \beta_{\text{LUT}}, \qquad \text{NAND2}(w) \approx \alpha_{\text{N}} \cdot L \cdot w + \beta_{\text{N}},$$
$$\text{FF}(w) \approx \alpha_{\text{FF}} \cdot L \cdot w + \beta_{\text{FF}}, \qquad P(w) \approx \alpha_P \cdot L \cdot w + \beta_P,$$

where the $\alpha$ coefficients capture per-node contributions (one LUT primitive, one pipeline flip-flop on the inter-layer boundary in max-throughput mode, the dynamic-power cost of its switching activity) and the $\beta$ terms collect the fixed-cost encoder comparator chain, head popcount/argmax tree, and device static-power floor. The timing path, in contrast, is set by layer depth $L$ and the head's $O(\log c)$ popcount reduction, *not* by $w$, because each LUT's $n$ inputs are evaluated independently and its gate delay does not grow with the number of sibling LUTs in the same layer. Accordingly $F_{\max}$, single-sample latency $C \cdot T$, and throughput $F_{\max}/$II are width-invariant to first order, so long as the design fits within the target part. Energy per sample composes linearly in the same way:

$$E(w) = \frac{10^6 \cdot P(w)}{F_{\max}/\text{II}} \approx \frac{10^6 \cdot \text{II} \cdot (\alpha_P \cdot L \cdot w + \beta_P)}{F_{\max}} \quad [\text{nJ/sample}].$$

Empirical support: the rank-2 node sweep underlying Figure 1a reports NAND2-equivalent gate count at $w \in \{0.5, 1, 2, 4, 8, 16\}$ K for LightLUT and recovers a near-linear fit. The MNIST-winner resource numbers of Tables 12 and 13 should therefore be read as one point on this line. The timing numbers of Table 7 carry over unchanged for any $w$ that the target part can host.

### C.6 External reference rows: sources and caveats

**Sources.** The two FINN-R MLP-4 rows ($W^1A^1$, MNIST, 97.69% top-1) take clock and power from Table 5 of Blott et al. (2018). Samples per second is the reported GOp/s divided by 6.0 M ops per frame (their Table 4), and energy per sample is power divided by samples per second. The two MLPerf-Tiny IC rows take latency and energy per inference directly from Table 5 of Borras et al. (2022), converted to ns and nJ. Cells the source paper does not report ($F_{\max}$ for MLPerf-Tiny, latency for FINN-R) stay placeholders.

**Why they are not directly comparable.** None of the four rows are a like-for-like comparison with ours. The FINN-R rows run a binarised MLP and the MLPerf-Tiny rows run CIFAR-10 Convolutional Neural Networks (CNNs), so the model class differs in every row, and the dataset also differs in the MLPerf-Tiny rows. Our numbers are Vivado post-route static estimates, the FINN-R numbers are also predicted rather than measured, and only the MLPerf-Tiny rows are measured on real hardware. The reference power figures are board-level and include the ARM processing system and peripheral rails, while our U55C and XCZU7EV power numbers are device-level. The rows give an order-of-magnitude reference, not a head-to-head comparison.

