# OpenReview forum: "BitLogic: A Framework for Gradient-Based LUT-Native Neural Networks"
_TMLR — Accepted by TMLR_

### Review · Reviewer_ngri · 2026-03-18

**Summary Of Contributions:**

The paper introduces BitLogic, a framework for training FPGA-native neural networks using differentiable lookup-table (LUT) nodes. The framework enables gradient-based training of discrete logic models and supports automatic export of trained PyTorch models to synthesizable FPGA hardware. The authors evaluate the framework on several vision benchmarks and report both predictive performance and hardware efficiency across multiple platforms.

Strengths:
1. This paper presents a well-structured framework BitLogic that unifies several recent approaches to LUT-based neural networks and provides a modular infrastructure for experimentation.
2. This paper includes an end-to-end pipeline from training to hardware deployment, including automatic RTL export and FPGA synthesis, which is valuable for reproducibility and practical deployment.
3. This paper provides systematic component-level analyses and hardware profiling, which help clarify the impact of different design choices in LUT-based models.

Weaknesses:
1. The novelty is limited. The main contribution is primarily a framework and system integration effort, while most of the core modeling components are adapted from prior work.
2. Evaluation scope is limited. The evaluation focuses on relatively small image classification benchmarks (e.g., MNIST, Fashion-MNIST, CIFAR). It would strengthen the work to evaluate it on a broader range of tasks or architectures (e.g., larger vision models, sequence modeling tasks)
3. Some architectural configurations appear under-explored. The convolutional variants perform substantially worse than the feedforward variants, and the paper suggests that these models were not extensively tuned. Additional experiments or architectural exploration could help clarify whether this gap reflects a limitation of the framework or simply insufficient tuning.

**Audience:**

Yes

**Audience Explanation:**

The work may be of interest to researchers studying hardware-aware machine learning, FPGA-based inference, and logic-based neural networks. In particular, the proposed framework and training pipeline could be useful for those exploring LUT-based neural architectures and hardware–software co-design.

**Claims And Evidence:**

No

**Claims Explanation:**

The paper generally provides empirical evidence that supports most claims.

However, the strength of the empirical evidence is somewhat limited. The evaluation focuses mainly on small image classification benchmarks, and comparisons with prior FPGA or logic-based approaches are difficult to interpret due to differences in hardware setups and reporting conventions. In addition, some architectural variants (e.g., the convolutional models) appear under-optimized, which makes it harder to fully assess the framework’s potential performance.

**Requested Changes:**

1. Strengthen the empirical evaluation. Including additional datasets or tasks, or providing more extensive comparisons with recent LUT-based or logic-based neural architectures, would help better demonstrate the generality and competitiveness of the framework.
2. Improve the evaluation of convolutional architectures. Additional tuning or analysis could clarify whether the performance gap is due to architectural limitations or experimental settings.
3. Provide clearer hardware comparison methodology. Since FPGA evaluation setups can vary significantly across studies, adding more standardized comparisons (e.g., normalized metrics or clearer synthesis assumptions) would make the reported hardware efficiency gains easier to interpret.

---

> ### Author Response · Authors · 2026-04-25
> **Cross-method comparison added; scope narrowed to feedforward image classification; hardware methodology unified.**
>
> Thank you for the balanced review. The revision is a substantial rewrite that addresses each weakness directly; we summarise the changes below.
>
> **Summary of changes.**
>
> - **Novelty reframed.** §1 no longer claims novel relaxations, layers, or heads. The contributions are now (i) the BitLogic framework with all five axes — encoder, connectivity, fan-in, node parameterization, head — independently swappable; (ii) a protocol-matched cross-method comparison built from per-axis sweeps that identify a new best-of-space configuration plus six gradient-trained priors retrained on four image-classification benchmarks; and (iii) a unified GPU + FPGA + ASIC evaluation of the resulting checkpoint from one path, with bit-exact Python and deployed-hardware accuracy.
> - **Empirical scope broadened.** §6 retrains six gradient-trained priors inside BitLogic on MNIST, Fashion-MNIST, CIFAR-10, and CIFAR-100 under one shared protocol; the best-of-space row wins 62 of 72 (dataset, width, prior) cells.
> - **Convolutional variants removed.** Empirical claims are scoped to feedforward two-layer image classification (§1 Scope, §8); convolutional, residual, attention, and recurrent extensions are out of the main paper.
> - **Single hardware methodology.** §7 uses NAND2-equivalent gate count (Yosys + Nangate 45 nm) as the target-independent primary metric and Vivado post-route LUT count on Alveo U55C and Zynq UltraScale+ XCZU7EV as deployment reference; encoder and head are included in every reported number; bit-exact Python-to-HDL agreement is verified by Verilator (§4 Evaluation, §C).
>
> **1. Limited novelty.** The paper no longer claims novelty for relaxations, layers, or heads. §3 ("The five-axis design space") recasts feedforward LUT networks as five interchangeable axes — encoder, connectivity, fan-in, node parameterization, head — and Table 8 maps every prior method to a point in that space. The contribution list in §1 is now: (i) the BitLogic framework, released so subsequent work can evaluate a new axis against the whole published slate, with all five axes independently swappable; (ii) a protocol-matched cross-method comparison combining per-axis sweeps that identify a new best-of-space model with six gradient-trained priors retrained on four benchmarks; and (iii) a unified GPU + FPGA + ASIC evaluation of the resulting checkpoint from one path, with Python and deployed-hardware accuracy bit-exact by construction.
>
> **2. Empirical scope.** §6 retrains six gradient-trained priors (DiffLogic, PolyLUT, NeuraLUT, DWN, WARP-LUT, LILogicNet) inside BitLogic under one shared protocol on MNIST, Fashion-MNIST, CIFAR-10, and CIFAR-100. The best-of-space row wins 62 of 72 (dataset, width, prior) cells; the remaining 10 are DWN out-of-memory entries (DWN's $O(w^{2})$ fast-path and 10-bit thermometer exceed the training-GPU budget). At $w=64\text{K}$, headline accuracies are $97.84\,/\,89.16\,/\,58.06\,/\,18.82\,\%$ on MNIST / Fashion-MNIST / CIFAR-10 / CIFAR-100. Non-classification tasks and depth are flagged out of scope and revisited in §8 ("Depth, the open sixth axis"; "Extensibility").
>
> **3. Convolutional underperformance.** Convolutional, residual, attention, and recurrent extensions are removed from the main paper; empirical claims are scoped to feedforward two-layer image classification (§1 Scope; §8). The five-axis sweep transfers to those topologies by construction once depth and topology are added as further axes (§8 Outlook).
>
> **4. Hardware comparison methodology.** §7 follows one protocol throughout: NAND2-equivalent gate count from Yosys + Nangate 45 nm as the target-independent primary metric (used in every sweep table of §5 and in §6), Vivado post-route LUT count on Alveo U55C and Zynq UltraScale+ XCZU7EV as the deployment reference, encoder and head included in every reported number, three emission modes per FPGA target (max throughput, lowest latency, fewest resources; Table 7), and bit-exact agreement between the Python forward path and the emitted SystemVerilog (§4 Evaluation, §C). Published numbers from prior works are clearly labelled — a "Published" column in Table 6 and a separate row group in Table 7 — and explicitly framed as not directly comparable to our protocol-matched rows.

---

### Review · Reviewer_uD6y · 2026-04-06

**Summary Of Contributions:**

This paper presents BitLogic, a gradient-based framework for FPGA-native neural networks based on LUT computation, together with an automatic PyTorch-to-RTL export pipeline. The paper also studies several architectural components and reports accuracy and hardware results on image-classification benchmarks like CIFAR-10.

**Additional Comments:**

1. Abstract: "FPGA provide an attractive substrate" should be "FPGAs provide an attractive substrate."

2. Sec. 2.2: "Guo (2025) survey LUT-based FPGA DNNs" should be "surveys."

3. Sec. 2.3: "logic operatored based components" is ungrammatical and should be revised.

**Audience:**

Yes

**Audience Explanation:**

Researchers working on FPGA-based ML acceleration, hardware-aware neural networks, and LUT or logic-based learning models are likely to be interested in this paper.

**Claims And Evidence:**

No

**Claims Explanation:**

1. The paper makes broad claims about supporting arbitrary models across diverse domains with automatic RTL generation, but the actual evaluation is limited to four image-classification datasets and two architecture families. The paper also states in Sec. 5.1 that the current implementation mainly focuses on image classification and that HDL export is robust mainly for feedforward, fully combinational graphs, so the claimed generality is not fully demonstrated.

2. The comparison with prior work is not broad enough for the paper's overall framing. Table 1 mainly compares against logic-based neural baselines, while a stronger discussion is needed against the broader FPGA DNN literature, including works such as the DAC SDC FPGA solutions (https://pku-sec-lab.github.io/dac-sdc-2024/results-fpga/) and recent LUT-assisted multiplier designs such as LUTMUL (https://arxiv.org/abs/2411.11852).

3. The ablation does not fully support some of the claimed architectural contributions. In Table 2, GroupedDSP performs worse than GroupSum, which weakens the claim that it is a practically better design choice, and several other highlighted components such as attention, residual connections, and transposed convolutions are not clearly validated in the main experiments.

4. The hardware evidence is also less convincing than the presentation suggests. In Table 3, the FPGA result is reported for a reduced model because of Vivado synthesis limitations, and the FPGA latency and energy are based on post-synthesis estimates rather than direct deployment measurements, which makes the comparison less direct.

**Requested Changes:**

1. Narrow the scope of the claims, or expand the empirical validation so it better matches the claims about arbitrary models, diverse domains, and automatic RTL generation.

2. Add a more complete comparison or discussion against broader FPGA DNN accelerators, not only logic-based neural baselines, so readers can better understand where BitLogic stands relative to existing FPGA implementations.

3. Revise the discussion of the proposed components so that it matches the ablation evidence. In particular, the GroupedDSP claim should be toned down or supported better, and the contributions of attention, residual connections, and transposed convolutions should be validated more directly if they are kept as major claims.

4. Clarify the hardware evaluation setup in the main text, especially that the FPGA result uses a reduced model due to synthesis limitations and that the FPGA latency and energy numbers come from post-synthesis estimates rather than direct deployment measurements.

---

> ### Author Response · Authors · 2026-04-25
> **Claims narrowed, FPGA-DNN context added, unsupported components dropped, hardware caveats made first-class.**
>
> Thank you for the targeted critique. The tightening of claims, the broader literature context, and the hardware caveats you asked for are all reflected in the rewrite.
>
> **1. Scope of claims vs. evaluation.** The abstract, §1, and contribution list are rewritten around a narrow claim: a five-axis design-space view of feedforward LUT networks, a per-axis sweep that synthesizes a new best-of-space model, the first protocol-matched cross-method comparison, and the first unified GPU + FPGA + ASIC hardware report. "Arbitrary models across diverse domains" is gone; the Scope paragraph in §1 and §8 restrict empirical claims to **feedforward two-layer image classification**. Convolutional, attention, residual, and transposed-conv blocks are removed from the main paper.
>
> **2. Broader FPGA-DNN comparison.** §2 ("Orthogonal and complementary directions") positions BitLogic against the broader FPGA-DNN line: HLS4ML (Duarte 2018), FINN / FINN-R (Umuroglu 2017; Blott 2018), LUTMUL (Xie 2025), the LUT-DNN survey (Guo 2025), TreeLUT (Khataei 2025), LL-ViT (Nag 2025), ConvDiffLogic (Petersen 2024), and Kresse (2025). Because the underlying primitive in those works is arithmetic — or the target is a specific application benchmark such as DAC-SDC — apples-to-apples accuracy comparisons are explicitly framed as out of scope.
>
> **3. Ablation vs. component claims.** GroupedDSP is no longer a proposed contribution. It appears as one option in the head sweep (§5.5, Table 5) alongside the popcount head; the takeaway states that GroupSum is competitive at zero DSPs and that GroupedDSP is "essentially unjustified at $w=32\text{K}$". The new best-of-space configuration uses the popcount head. Attention, residual, transposed-conv, and convolutional heads and layers are removed from the main paper, so there is no claim/evidence mismatch.
>
> **4. Hardware evaluation clarity.** Two caveats are now first-class in §7 and §C: the deployed cell is the MNIST best-of-space winner at $w=4{,}000$ — the widest cell the Vivado flow admits within our build-host memory budget; wider cells extrapolate linearly in resource and power at unchanged timing (§C). FPGA $F_\mathrm{max}$ comes from Vivado post-route timing, latency and throughput follow from $F_\mathrm{max}$ and II, and power is from Vivado's vectorless activity estimation on the full-module routed netlist (post-route estimate, not on-board measurement; flagged in §7 and as future work in §8). The case study spans two FPGA targets (Alveo U55C, Zynq UltraScale+ XCZU7EV), three emission modes per target, three GPU references at two batch sizes, and an ASIC proxy via Yosys + Nangate 45 nm. Per-submodule resource breakdowns are in Tables 11–13. The Python forward path is bit-exact with the emitted SystemVerilog (Verilator-checked on the full MNIST test set), so every accuracy number is the deployed-network accuracy.
>
> **5. Typos.** All three are fixed; the offending phrasings do not appear in the rewritten text.

---

### Review · Reviewer_ULok · 2026-04-12

**Summary Of Contributions:**

The paper presents **BitLogic**, a framework for building, training, and exporting LUT/logic-based neural networks for FPGA deployment. The submission aims to unify a previous literature in which prior works often use different codebases, training procedures, preprocessing assumptions, and hardware reporting conventions, making fair comparison difficult. The framework is modular, supporting multiple node relaxations, layer/connectivity schemes, encoders, heads, and an RTL export pipeline.

The paper’s main strength is its **motivation**. A common framework for LUT-/logic-based neural models could be genuinely useful to the community, especially if it enables reproducible and end-to-end comparison across methods under shared hardware assumptions.

However, the paper has several important weaknesses. First, the **contributions and experiments are narratively unclear**: the paper is presented as a framework/infrastructure paper, but it is in methodology and experiments as though it were introducing novel architectural components, even though the design space is drawn from prior work. Second, the **core promise of fair comparison is not actually demonstrated** in the experiments: the paper compares against results reported by prior papers rather than using the proposed framework to retrain/export representative baselines under common assumptions. Third, the **hardware evaluation is not sufficient**, especially the gate-count comparison, which is computed in a non-standard fashion, computed differently across architectures, and excludes encoder and head costs.

**Audience:**

Yes

**Audience Explanation:**

Yes. I believe at least a meaningful subset of the TMLR audience would be interested in this paper’s topic and goals.

There is clear interest in hardware-aware machine learning, efficient inference, FPGA deployment, quantized/binary models, and methods that bridge ML with digital design flows. Researchers working on LUT-based neural networks, differentiable logic models, neural architecture/hardware co-design, and deployment-oriented ML systems would likely find the problem this paper addresses relevant. Even readers outside this exact niche may be interested in the broader question of how to build reusable infrastructure that makes specialized model classes easier to compare and reproduce.

That said, my positive answer here reflects the **importance of the problem and potential relevance of the framework idea**, not a judgment that the current paper has already delivered that contribution. I do think the paper’s goals are of interest to part of the TMLR audience, but I also think the work needs substantial revision before it is demonstrated convincingly enough for acceptance.

**Broader Impact Concerns:**

I do not have major broader-impact concerns that would require a dedicated statement beyond standard discussion of efficient hardware deployment.

**Claims And Evidence:**

No

**Claims Explanation:**

I do not think the paper’s main claims are fully supported by accurate, convincing, and clear evidence in its current form.

The central claim of the paper is that BitLogic provides a unified framework that enables fair and reproducible comparison across LUT-/logic-based neural models targeting FPGA deployment. However, the experimental section does not actually demonstrate such a unified comparison. Instead, the paper mainly compares its own instantiated models against results reported in prior publications. Those prior results come from different codebases, preprocessing pipelines, training budgets, architectural assumptions, and hardware evaluation methodologies. This is precisely the comparability problem that the paper identifies in the introduction. As a result, the experiments do not convincingly validate the paper’s contribution as a framework.

A second issue is the paper’s treatment of **novelty**. The submission claims novel architectural and training components, but the methods described in Sections 3.1-3.3 and the appendix are implementations and reformulations of existing ideas rather than new contributions. The paper should present these sections as background instead of implying broader methodological novelty than is actually established.

A third problem is the **hardware evidence**, especially the ``gate count'' comparisons. The paper explicitly excludes encoder and classification-head costs from the BitLogic counts, even though these are part of the deployed inference pipeline and is significant. In addition, counts for prior works are taken from their original papers using heterogeneous conventions. The resulting table therefore combines quantities that are not directly comparable. This weakens the paper’s claims about hardware efficiency and fair evaluation. The convolutional case is even less clear, since the counts do not adequately specify whether they refer to folded/shared execution or fully unfolded replicated implementations, which has major consequences for area and latency.

The FPGA evaluation is also too limited for a framework paper. The paper reports essentially one FPGA case study, and even there the implemented FPGA configuration is reduced relative to the larger software model because of synthesis/tool limitations. This is useful as a proof of concept for export, but it does not convincingly demonstrate the broader claim that the framework enables fair cross-model hardware comparison.

For these reasons, I do not think the current submission provides sufficiently strong evidence for its main claims, even though I think the underlying idea is worthwhile and has potential.

**Requested Changes:**

### Critical changes

1. **Reframe the contribution more clearly**

   The paper should clearly state whether its primary contribution is:
   - a new model family,
   - a framework/infrastructure contribution,
   - or a benchmarking/evaluation methodology.

   In my view, the paper is strongest as a **framework/infrastructure paper**, but the current draft partly frames itself as introducing multiple novel architectural components. The authors should revise the narrative so that inherited and reimplemented ideas are clearly distinguished from any genuinely new technical contributions. Sections 3.1--3.3 should largely be treated as background/framework design unless the authors can clearly justify specific novelties.

2. **Demonstrate the claimed unified comparison using the proposed framework**

   This is the most important change. The paper argues that prior work is hard to compare fairly because each method uses different code and hardware assumptions. Therefore, the paper should actually use BitLogic to instantiate, train, and export at least a representative subset of prior LUT-/logic-based approaches under a shared protocol, reporting metrics such as LUT utilization, latency, Fmax, throughput, power, etc.

3. **Replace or substantially revise the gate-count comparison**

   The current gate-count table is not sufficiently meaningful for the claims being made. The revised paper should:
- Use their proposed framework to reproduce prior work
   - include encoder/preprocessing cost (thermometer / quantization).
   - include classification head and final class-selection cost
   - unfold convolutions in gate count
   - Ideally, the paper should report common post-synthesis hardware metrics through NAND2-equivalent estimates.

4. **Strengthen the FPGA/hardware evaluation**

   A framework paper should not rely on essentially a single reduced-size FPGA example. The authors should provide a broader hardware evaluation that demonstrates the framework’s comparative value, e.g. multiple representative architectures exported and evaluated on the same target.

5. **Improve reproducibility**

   For a framework paper, reproducibility is central. The revision should provide:
   - exact configs for reported results,
   - scripts for reproducing each single result. This is currently not provided in supplementary material code.

6.  **Historical Context and Baseline Definitions:**

- LUT Origins: The paper claims the shift toward LUT-based NNs began with LUTNet in 2019. This ignores decades of weightless neural network literature, such as n-tuple networks and WiSARD. This history must be acknowledged.
- DiffLogic Mischaracterization: The authors frame differentiable logic gate networks (e.g., DiffLogic) as a "distinct paradigm" from LUTs. In reality, a 2-input logic gate is mathematically and functionally identical to a 2-input LUT ($n=2$). The hypothesis space is the same.

7. **Improve attribution and citations**

   - Several citations are missing the paper, especially in the encoder. Thermometer, Gaussian Thermometer, Distributive Thermometer, all have their respective papers.

8. **Table 2**

    Table 2 presents an ablation study comparing different components while fixing the base model to a two-layer FFN with 4,000 units per layer. This methodology is highly misleading, because different components can have very different optimal architectures. Evaluating all components under the same architecture can therefore misrepresent which component performs best.

    To properly ablate different components, the best architecture for each component should first be identified through hyperparameter search, under a fixed hardware budget (for example, LUT count or latency) or under an iso-accuracy setting. The components should then be compared fairly: by accuracy when hardware is fixed, or by hardware metrics when accuracy is fixed. I recommend training for iso-accuracy and reporting FPGA metrics for comparison.

---

> ### Author Response · Authors · 2026-04-25
> **Reframed as design-space + cross-method comparison; gate-count and hardware evaluation overhauled.**
>
> Thank you for the detailed critique. The revision restructures the contribution, methodology, and hardware evaluation around your critical changes.
>
> **1. Reframe the contribution.** The paper is now a five-axis design-space view of feedforward LUT / logic-gate networks plus a protocol-matched cross-method comparison enabled by the BitLogic framework. §1 names three contributions in this order: the released framework with all five axes independently swappable; the protocol-matched cross-method comparison combining per-axis sweeps and six retrained priors; and a unified GPU + FPGA + ASIC evaluation from one path. §3 treats every prior method as a point in the space and Table 8 maps the nine prior methods (LUTNet, LogicNets, DiffLogic, LightLUT, PolyLUT, NeuraLUT, DWN, WARP-LUT, LILogicNet) to their BitLogic coordinates. The relaxations, layers, and heads of the prior body of work are framed as background; we no longer claim novelty for them.
>
> **2. Unified comparison inside BitLogic.** §6 retrains six gradient-trained priors (DiffLogic, PolyLUT, NeuraLUT, DWN, WARP-LUT, LILogicNet) under one shared protocol (AdamW, $\eta=0.01$, batch 128, 100 epochs) at $w \in \{4\text{K}, 16\text{K}, 64\text{K}\}$, two seeds, on MNIST, Fashion-MNIST, CIFAR-10, and CIFAR-100. The best-of-space row wins **62 of 72** cells; the 10 misses are DWN out-of-memory entries (its $O(w^{2})$ fast-path plus 10-bit thermometer exceeds the training-GPU budget). At $w=64\text{K}$ the headline accuracies are $97.84\,/\,89.16\,/\,58.06\,/\,18.82\,\%$. §6 and §A are explicit that the shared protocol controls the training recipe but not each method's bespoke pipeline (PolyLUT structured pruning, NeuraLUT-Assemble, DWN calibration, WARP-LUT residual block); §A lists those pipelines as deliberately disabled, and the gap to the "Published" column is attributed to protocol- and pipeline-effects rather than to design-space coordinates.
>
> **3. Gate-count methodology.** Overhauled to your specification. NAND2-equivalent count from Yosys + Nangate 45 nm is the primary metric in every sweep table of §5 and in §6. Encoder and head are now included in every NAND2-eq and Vivado LUT number — the previous encoder-exclusion is gone. Vivado post-route LUT count is retained as the deployment reference on Alveo U55C and Zynq UltraScale+ XCZU7EV (§7). The $2^{n}-1$ proxy is removed from every reported metric; it survives only as a worst-case scaling note in §3. Convolution unfolding no longer applies — the paper is feedforward-only.
>
> **4. Broader hardware evaluation.** §7 emits one MNIST best-of-space checkpoint to three backends: a bit-packed GPU path, Vivado post-route on two FPGAs, and a Yosys + Nangate 45 nm ASIC proxy. Each FPGA target is reported in three emission modes (max throughput, lowest latency, fewest resources), alongside three consumer GPUs at two batch sizes. Tables 7 and 11–13 report $F_\mathrm{max}$, latency, throughput, power, energy/sample, Vivado LUT/FF, and NAND2-eq with per-submodule breakdowns. At $88.79\,\%$ on MNIST ($w=4{,}000$ — the widest cell within the build-host memory budget) the FPGA reaches $126.6$ / $127.2$ MSamp/s on U55C / XCZU7EV — about $15\times$ the RTX 3090 throughput at four to five orders of magnitude less energy ($6$–$27$ nJ/sample, max-throughput mode). FPGA numbers are post-route estimates; on-device measurement is flagged as future work in §8.
>
> **5. Reproducibility.** Every numerical result in §5–§7 maps to a single released configuration; the paper-to-config mapping is in §A. Dependencies are version-pinned; code is released on acceptance.
>
> **6. Historical context.** §2 ("Weightless neural networks") opens with the n-tuple / WiSARD / BTHOWeN lineage (Bledsoe & Browning 1959; Aleksander et al. 1984; Susskind et al. 2022) and notes that modern differentiable-LUT methods revisit this idea with gradient-based training. The paper no longer implies the field began with LUTNet.
>
> **7. DiffLogic ≡ LUT hypothesis space.** §3 ("Nodes are parameterizations, not paradigms") states explicitly that every node parameterization discretizes to the same $2^{2^{n}}$ Boolean truth table at deployment, so the deployed format is identical across the field; differences are in optimization dynamics and parameter count, not in representable functions. §6 treats DiffLogic as one node choice on equal footing with the others.
>
> **8. Encoder citations.** §3 cites Buckman et al. (2018) for the linear thermometer and Bacellar et al. (2022) for the distributive (quantile) thermometer; both are in the bibliography and cited where each encoder family appears.
>
> **9. Ablation methodology.** Per-axis sweeps are run at three widths each (Tables 1–5), separating axis signal from width scaling. §8 explicitly flags the limitation that per-axis sweeps hold the base architecture fixed. The §6 cross-method contest then combines the per-axis winners against each prior method's own coordinates.

---

### Decision · Action_Editor_B8Vm · 2026-05-26

**Recommendation:** Accept with minor revision

**Additional Comments:**

- State in the Abstract that evaluations are limited to two-layer feed-forward networks.
- State in the Abstract that method-specific procedures such as calibration, pruning, thresholding are not considered in the framework.

- Tables are not referenced in the main text. Reference tables appropriately.
- Formula (1): The semantics of the formula is unclear. Should be described. Define d and b.
- paragraph "Connectivity is the main per-layer design choice": define j in M_j and w_in.
- Start of Section 4 and later: What is meant with "body"?
- Sec 4, paragraph "Architecture": (1) is not an equation.
- Replace arXiv or workshop versions of referenced papers by journal versions where available.

**Audience:**

Yes

**Audience Explanation:**

The manuscript is of interest for the field of hardware-aware and efficient machine learning.

**Claims And Evidence:**

Yes

**Claims Explanation:**

The main claim of the submission is that they provide a framework that enables to instantiate every prior method for gradient-based LUT networks under one shared training and evaluation protocol.
The authors provide such a framework, which they call BitLogic. BitLogic has limitations which are reasonably well stated. I propose to further clarify these limitations in the Abstract.